# Simple Graph Contrastive Learning via Fractional-order Neural Diffusion Networks

## Abstract

Graph Contrastive Learning (GCL) has recently made progress as an unsupervised graph representation learning paradigm. GCL approaches can be categorized into augmentation-based and augmentation-free methods. The former relies on complex data augmentations, while the latter depends on encoders that can generate distinct views of the same input. Both approaches may require negative samples for training. In this paper, we introduce a novel augmentation-free GCL framework based on *graph neural diffusion models*. Specifically, we utilize learnable encoders governed by Fractional Differential Equations (FDE). Each FDE is characterized by an order parameter of the differential operator. We demonstrate that varying these parameters allows us to produce learnable encoders that generate diverse views, capturing either local or global information, for contrastive learning. Our model does not require negative samples for training and is applicable to both homophilic and heterophilic datasets. We demonstrate its effectiveness across various datasets, achieving state-of-the-art performance.

## 1. Introduction

Contrastive learning is a powerful unsupervised learning technique that has gained significant attention in representation learning. It focuses on learning meaningful representations by distinguishing between similar and dissimilar feature embeddings generated from different *encoders*. The learning process pulls similar instances closer together while pushing dissimilar ones apart in the feature space. This approach enables models to capture important patterns without requiring extensive labeled data. Contrastive learning has been widely applied in areas such as computer vision, natu-

ral language processing, and recommender systems. When this technique is applied to unsupervised learning involving graph-structured data, it is known as *Graph Contrastive Learning (GCL)*.

GCL methods can be categorized into *augmentation-based* and *augmentation-free* approaches (see Section 2). Augmentation-based methods rely on complex data augmentations, while augmentation-free methods depend on encoders that can generate distinct views of the same input. Both approaches may require negative samples for training. We focus on the augmentation-free approach due to its simplicity and independence from the quality of augmentations. However, the success of an augmentation-free approach hinges on two factors: (a) the ability of the encoders to generate high-quality feature embeddings, and (b) the capability of contrasting encoders to produce distinct views of the same input. To address these requirements, we propose a novel GCL framework that utilizes neural diffusion-based encoders to generate contrasting views of node features.

To explain our main insight regarding applying neural diffusion in GCL, recall that recent works Chamberlain et al. (2021a;b); Kang et al. (2023); Rusch et al. (2022); Song et al. (2022); Thorpe et al. (2022) have introduced graph diffusion based on ordinary differential equations (ODE). This approach is analogous to heat diffusion over a graph and can be viewed as a continuous substitution for the message passing of node features in models such as GCN. Diffusions based on *fractional-order differential equations (FDE)* with the differential operator $\frac{\mathrm{d}^\alpha}{\mathrm{d}t^\alpha}$, which generalize ODE-based diffusions, have been proposed in Kang et al. (2024); Zhao et al. (2024). The key parameter is the order $\alpha \in (0, 1]$ of the derivative of features with respect to (w.r.t.) time $t$. For instance, $\alpha = 1$ corresponds to the usual derivative $\frac{\mathrm{d}}{\mathrm{d}t}$. FDE allows $\alpha$ to be a real number, e.g., $0 < \alpha < 1$, with the interpretation that it governs how much "non-local" information (from the past feature evolution history) is incorporated in the diffusion. Therefore, we can adjust $\alpha$ to control whether the features contain more global or local information. By choosing different $\alpha$ values for different FDE-based encoders, we can generate features with distinct views, which is essential for augmentation-free GCL.

The term *diffusion* has been associated with different mean-

[1]Anonymous Institution, Anonymous City, Anonymous Region, Anonymous Country. Correspondence to: Anonymous Author <anon.email@domain.com>.

Preliminary work. Under review by the International Conference on Machine Learning (ICML). Do not distribute.

ings depending on the context. In our paper, it specifically refers to the dynamic of the features following a prescribed differential equation. Our main contributions are as follows:

- We introduce a novel augmentation-free GCL model utilizing neural diffusion-based encoders on graphs. This model is both simple and flexible, with the feature view properties primarily governed by a single parameter within the continuous domain $(0, 1]$.

- We provide a theoretical analysis of the model using the framework of FDE-based graph neural diffusion models, offering insights into the features generated by contrasting encoders. Based on the observations, we propose a new way to regularize contrastive loss, avoiding the use of negative samples.

- We conduct extensive numerical experiments on datasets with varying characteristics (e.g., homophilic vs. heterophilic) and demonstrate the model's superior performance compared to existing benchmarks.

## 2. Related Works

In this section, we summarize recent advancements in GCL that are related to our work. Additional details on these methods can be found in Appendix B.

*Augmentation-based* GCL models typically use various data augmentations, such as edge removal and node feature masking, to create diverse graph views. These models optimize the loss by maximizing mutual information between the augmented views (Chen & Kou, 2023; Chen et al., 2022; Hassani & Khasahmadi, 2020a; Velickovic et al., 2019; Xiao et al., 2022; You et al., 2020; Zhang et al., 2023; Zhu et al., 2020; 2021). This approach enhances model robustness and generalization.

In contrast, *augmentation-free* methods do not rely on complex augmentations. They directly input the same graph and features into different encoders to generate contrasting views (Peng et al., 2020; Zhang et al., 2022). Notably, methods such as GraphACL (Xiao et al., 2023) and SP-GCL (Wang et al., 2023) strictly avoid augmentations but still use negative samples. Similarly, PolyGCL (Chen et al., 2024) applies feature shuffling for negative samples. These approaches move away from the traditional homophily assumption, enabling more effective learning in heterophilic and structurally diverse graphs by incorporating advanced encoding and spectral filtering techniques.

A notable trend in GCL models is the elimination of negative sample pairs. For instance, BGRL (Thakoor et al., 2022) and CCA-SSG (Zhang et al., 2021) minimize the need for augmentations and completely remove negative sampling, focusing on maximizing correlations between graph views.

AFGRL (Lee et al., 2022) takes this further by eliminating both augmentations and negative samples, directly using the original graph to create positive samples. However, its applicability is primarily limited to homophilic datasets, posing challenges in handling heterophilic or structurally diverse graphs.

Our approach *neither requires any data augmentation nor negative sampling*, while effectively handling *both* homophilic and heterophilic datasets. Therefore, we regard our model with these characterizations as a "simple" form of GCL in the title. As highlighted in Section 1, unlike previous works, we use encoders based on *fractional-order* neural diffusion to generate pairs of feature representations for different views. As far as we are aware of, this is the first work along this line.

## 3. Preliminaries

**Setup and problem formulation**    Consider an undirected graph $\mathcal{G} = (\mathcal{V}, \mathcal{E})$, where $\mathcal{V} = \{v_1, \ldots, v_N\}$ is a finite set of $N$ nodes and $\mathcal{E} \subset \mathcal{V} \times \mathcal{V}$ is the set of edges. The raw features of the nodes are represented by the matrix $\mathbf{X}$, with the $i$-th *row* corresponding to the feature vector $\mathbf{x}_i$ of node $v_i$. The weighted symmetric adjacency matrix $\mathbf{A} = (a_{ij})$ has size $N \times N$, where $a_{ij}$ denotes the edge weight between nodes $v_i$ and $v_j$. We denote the complete graph information by $\mathcal{X} = (\mathbf{A}, \mathbf{X})$.

In unsupervised feature learning, we aim to learn an encoder $f_\theta$ that maps the raw feature $\mathbf{x}_i$ of each node $v_i$ to a refined feature representation $\mathbf{z}_i = f_\theta(\mathcal{X}, v_i)$ in $\mathbb{R}^F$. The resulting encoded node features $\mathbf{Z} \in \mathbb{R}^{N \times F}$, where each row corresponds to $\mathbf{z}_i$, are then utilized for downstream tasks such as node classification, which is the primary focus of our paper. In GCL, self-supervision is achieved by encouraging consistency between features $\mathbf{Z}_1$ and $\mathbf{Z}_2$ generated from distinct encoders $f_{\theta_1}$ and $f_{\theta_2}$.

**Graph neural diffusion models**    Traditional Graph Neural Network (GNN) models, such as GCN and GAT (Hamilton et al., 2017; Kipf & Welling, 2017; Veličković et al., 2018), rely on (discrete) graph message passing for feature aggregation. In the $k$-th iteration, the message passing step is represented as $\overline{\mathbf{A}}\mathbf{Z}^{(k-1)}$, where $\overline{\mathbf{A}}$ is the normalized adjacency matrix of $\mathbf{A}$, and $\mathbf{Z}^{(k-1)}$ is the output from the $(k-1)$-th iteration.

In contrast, Chamberlain et al. (2021b) has introduced a continuous analog to message passing, akin to heat diffusion in physics. The evolution of the learned feature $\mathbf{Z}(t)$ is governed by an ODE:

$$\frac{\mathrm{d}}{\mathrm{d}t}\mathbf{Z}(t) = \mathcal{F}(\mathbf{W}, \mathbf{Z}(t)), \qquad (1)$$

with the initial condition $\mathbf{Z}(0)$ being either $\mathbf{X}$ or its trans-

formed version. Here, $t$ represents the time parameter analogous to the layer index in GCN, while $\mathcal{F}(\mathbf{A}, \mathbf{Z}(t))$ denotes the spatial diffusion. A typical choice for $\mathcal{F}(\mathbf{W}, \mathbf{Z}(t))$ is $-(\mathbf{I} - \overline{\mathbf{A}})\mathbf{Z}(t)$, where $\mathbf{I}$ is the identity matrix.

This formulation is extended in Kang et al. (2024) by incorporating fractional-order derivatives. Specifically, for each *order parameter* $\alpha \in (0, 1)$, there exists a *fractional-order differential operator* $D_t^\alpha$ (see Appendix D). This operator generalizes $\frac{\mathrm{d}}{\mathrm{d}t}$ (in (1)) such that as $\alpha \to 1$, $D_t^\alpha$ converges to $\frac{\mathrm{d}}{\mathrm{d}t}$. Therefore, it is reasonable to set $D_t^1 = \frac{\mathrm{d}}{\mathrm{d}t}$, allowing $\alpha$ to be chosen from the interval $(0, 1]$.

Mimicking (1), once we fix $\alpha$, we obtain a new dynamic of the features $\mathbf{Z}(t)$ following an FDE as:

$$D_t^\alpha \mathbf{Z}(t) = \mathcal{F}(\mathbf{W}, \mathbf{Z}(t)). \tag{2}$$

The exact definition of $D_t^\alpha$ varies slightly in different contexts. However, they all share the common trait that $D_t^\alpha$ for $\alpha \in (0, 1)$ is defined by an integral. Intuitively, this implies that $\mathbf{Z}(t)$ depends on $\mathbf{Z}(t')$ for any $t' < t$. Therefore, unlike a solution to (1), the dynamic of $\mathbf{Z}(t)$ has "memory".

In our approach detailed in Section 4, by varying $\alpha$, we create a variety of distinct encoders, which are selected for augmentation-free GCL. In Appendices C and D, we provide exact definitions of $D_t^\alpha$ and discuss different variants of ODE- and FDE-based GNN models.

For a final remark, unlike the FLODE model (Maskey et al., 2023), which integrates fractional graph shift operators into graph neural ODEs to account for spatial domain rewiring and space-based long-range interactions during feature updates, the FDE model employs a time-fractional derivative to update graph node features, enabling a memory-inclusive process with time-based long-range interactions.

## 4. The Proposed Model: FD-GCL

In this section, we present our proposed **FDE D**iffusion-based **GCL** model, abbreviated as FD-GCL. We discuss the theoretical foundation of our approach in Section 4.1, providing insights on how different choices of the order $\alpha$ affect the view an encoder generates. Further analysis and numerical evidence are provided in Section 4.2. Full details of the FD-GCL model are presented in Section 4.3.

### 4.1. Encoders with Different Order Parameters

**FDE encoders** In theory, the evolution of node features $\mathbf{Z}(t)$ is governed by (2). For practical implementations, we adopt a *skip-connection mechanism* as described in Chamberlain et al. (2021b); Kang et al. (2024). This involves periodically adding the initial features to the embeddings after a fixed diffusion time $\tau$. To highlight the influence of the order parameter $\alpha$, we denote the resulting time-varying

features, incorporating skip-connections, as $\mathbf{Z}_\alpha(t)$.

If we use two encoders by solving FDEs with different orders $\alpha_1$ and $\alpha_2$, the resulting features are $\mathbf{Z}_{\alpha_1}(t)$ and $\mathbf{Z}_{\alpha_2}(t)$. As mentioned in Section 1, the effectiveness of GCL relies on whether $\mathbf{Z}_{\alpha_1}(t)$ and $\mathbf{Z}_{\alpha_2}(t)$ are high-quality feature representations with distinct views. Therefore, it is essential to analyze how the properties of $\mathbf{Z}_\alpha(t)$ vary with different choices of $\alpha$.

**Graph signal processing (GSP)** To present our main result, we need to introduce some concepts from graph signal processing (GSP) (Shuman et al., 2013). We provide a concise summary below. Consider the *normalized Laplacian* $\overline{\mathbf{L}} = \mathbf{I} - \overline{\mathbf{A}}$ (as used in the definition of $\mathcal{F}$ in (1)). Since $\overline{\mathbf{L}}$ is symmetric, it can be decomposed orthogonally as $\overline{\mathbf{L}} = \mathbf{U}\Lambda\mathbf{U}^\mathsf{T}$. In this decomposition, the diagonal entries $\lambda_1 \leq \ldots \leq \lambda_N$ of $\Lambda$ are the ordered eigenvalues of $\overline{\mathbf{L}}$ (also called the *graph frequencies*), and the $i$-th column $\mathbf{u}_i$ of $\mathbf{U}$ is the eigenvector corresponding to the eigenvalue $\lambda_i$. Each $\mathbf{u}_i$ represents a signal on the graph $\mathcal{G}$.

For small $\lambda_i$, $\mathbf{u}_i$ is smooth, meaning that signal values are similar across edges. Conversely, for large $\lambda_i$, $\mathbf{u}_i$ can be spiky, highlighting local features. Each signal $\mathbf{x}$ (e.g., a column of the feature matrix $\mathbf{X}$) can be expressed as a *spectral decomposition*:

$$\mathbf{x} = \sum_{i=1}^{N} c_i \mathbf{u}_i, \text{ where } c_i = \langle \mathbf{x}, \mathbf{u}_i \rangle.$$

In GSP, each $c_i$ is referred to as a *Fourier coefficient*, which measures the frequency response of $\mathbf{x}$ to the basis vector $\mathbf{u}_i$. For convenience, we say $\mathbf{x}$ has *large smooth components* if $|c_i|$ is relatively large for small indices $i$, and it is *energy concentrated* if $|c_i|$ is small for most indices $i$.

**The main result** We are now ready to state the main result using the concepts introduced above. A rigorous statement involves additional concepts and assumptions. To keep the discussion focused and avoid introducing unnecessary terms, we provide a detailed and rigorous explanation in Appendix E. We also explain the domain of $\alpha$ in Appendix D.

**Theorem 1** (Informal). *For $0 < \alpha_1 < \alpha_2 \leq 1$, the following hold for features $\mathbf{Z}_{\alpha_1}(t)$ and $\mathbf{Z}_{\alpha_2}(t)$ when $t$ is large:*

*(a)* $\mathbf{Z}_{\alpha_2}(t)$ *contains more large smooth components as compared with* $\mathbf{Z}_{\alpha_1}(t)$.

*(b)* $\mathbf{Z}_{\alpha_1}(t)$ *is less energy concentrated as compared with* $\mathbf{Z}_{\alpha_2}(t)$.

*Moreover, the contrast in (a) and (b) becomes more pronounced as the difference $\alpha_2 - \alpha_1$ increases.*

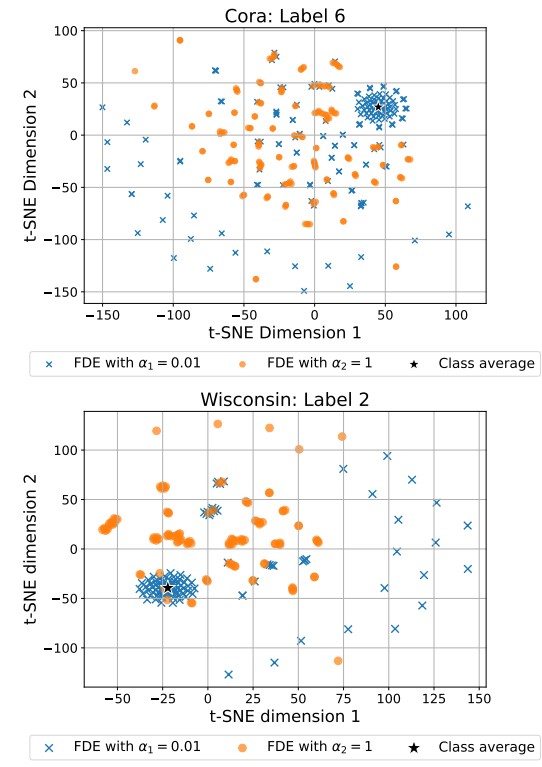

*Figure 1.* The t-SNE visualizations of node features from a single class, generated by encoders with different FDE order parameters. For comparison, features are linearly translated to align class averages. The datasets used are Cora (homophilic) and Wisconsin (heterophilic). The visualizations demonstrate that the two encoders produce embeddings with distinct characteristics. A smaller $\alpha$ produces features with a concentrated core, while features generated by a larger $\alpha$ are more evenly spaced. Additional results for other label classes are provided in Appendix G.

In the next subsection, we discuss the implications of Theorem 1 for our GCL model and provide insights into its strong performance as demonstrated in Section 5. We also present numerical evidence to support our claims whenever possible.

### 4.2. Contrasting Encoders

Following the setup in Section 4.1, we select $0 < \alpha_1 < \alpha_2 \leq 1$ as order parameters for FDE-based encoders, and obtain continuous sequences of features $\mathbf{Z}_{\alpha_1}(t)$ and $\mathbf{Z}_{\alpha_2}(t)$.

**Distinct views** As mentioned in Section 1, the success of an augmentation-free GCL hinges on whether different encoders generate feature representations with distinct views. We assert that $\mathbf{Z}_{\alpha_2}(t)$ encapsulates more global summary information, while $\mathbf{Z}_{\alpha_1}(t)$ captures finer local details.

More specifically, by Theorem 1(a), the smooth components (corresponding to small graph frequencies $\lambda_i$) play a pre-

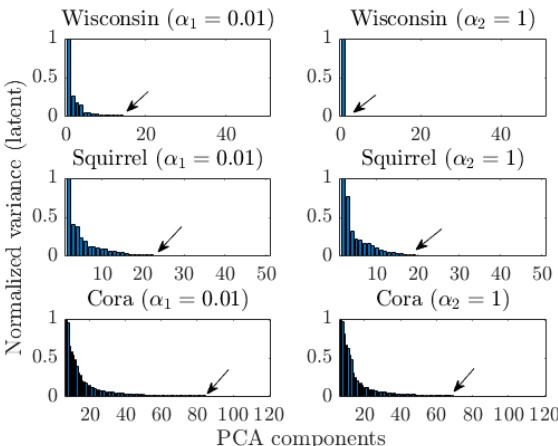

*Figure 2.* The PCA components of features for different datasets and choices of FDE order parameters. We see that for the small order ($\alpha_1$), the bar chart is comparatively more spread out, which prevents dimension collapse.

dominant role in $\mathbf{Z}_{\alpha_2}(t)$. Each such component is spanned by a vector $\mathbf{u}_i$, which is a signal with small variation across edges. Therefore, the global structural information is encoded in such a component. On the other hand, for $\mathbf{Z}_{\alpha_1}(t)$, non-smooth components $\mathbf{u}_i$, corresponding to large graph frequencies $\lambda_i$, have larger coefficients and hence contribution (as compared with $\mathbf{Z}_{\alpha_2}(t)$). As such a $\mathbf{u}_i$ is local in nature (spiky), $\mathbf{Z}_{\alpha_1}(t)$ captures local details as claimed. In summary, we have the correspondence: $\alpha_1 \longleftrightarrow$ "local", and $\alpha_2 \longleftrightarrow$ "global".

In Fig. 1, we numerically verify that the encoders generate different views by showing that the embeddings of $\mathbf{Z}_{\alpha_1}(t)$ and $\mathbf{Z}_{\alpha_2}(t)$ have distinct characterizations, on Cora (homophilic) and Wisconsin (heterophilic). Features generated with $\alpha_1$ form a concentrated core near the average, while there are deviated features. They can be counterbalanced by features generated with $\alpha_2$, which are more evenly spaced.

**Dimension collapse** *Dimension collapse* occurs when features are confined to a low-dimensional subspace within the full embedding space. This issue should be addressed and avoided in contrastive learning design. We assert that if $\alpha_1$ is small, then $\mathbf{Z}_{\alpha_1}(t)$ effectively avoids dimension collapse. According to Theorem 1(b), $\mathbf{Z}_{\alpha_1}(t)$ is less energy concentrated. This implies that the columns of $\mathbf{Z}_{\alpha_1}(t)$ can be represented as $\sum_{1 \leq i < N} c_i \mathbf{u}_i$ with a relatively bigger number of large $|c_i|$'s. Consequently, the features do not collapse into a low-dimensional space. As numerical evidence, we present the principal component analysis (PCA) decomposition of $\mathbf{Z}_{\alpha_1}(t)$ and $\mathbf{Z}_{\alpha_2}(t)$ for the Cora, Squirrel and Wisconsin datasets in Fig. 2. The results verify both Theorem 1(b) and our claim in this paragraph.

**Encoder quality**   The model's performance is expected to improve if the encoders can effectively cluster nodes of the same class independently. Our approach benefits from the proven performance of FDE-based encoders in supervised settings (Chamberlain et al., 2021b; Kang et al., 2024). We numerically verify their unsupervised clustering capability as follows: For each label class $c$, we compute the average intra-class feature distance $d_c^{\text{intra}}$ among nodes of class $c$ and the average inter-class feature distance $d_c^{\text{inter}}$ between nodes of class $c$ and nodes of other classes. The ratio $r_c = d_c^{\text{inter}}/d_c^{\text{intra}}$ serves as a measure of feature clustering quality, and the larger the better. The results, shown in Fig. 3, indicate that our proposed encoders generate high-quality feature embeddings.

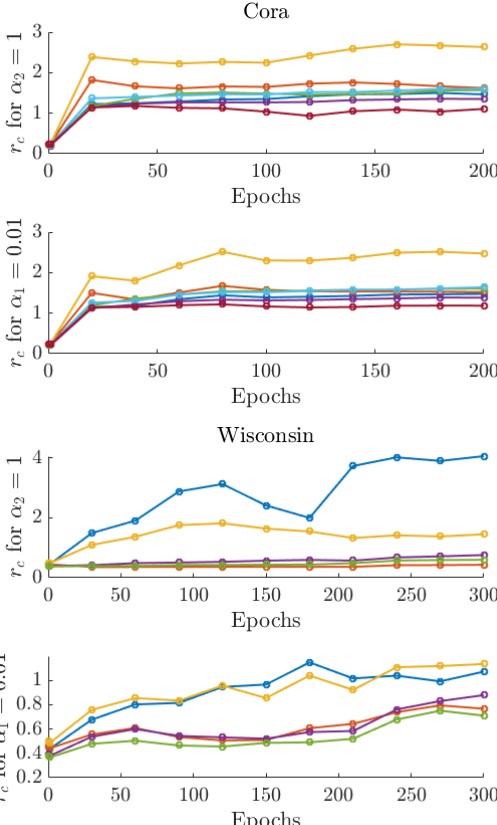

*Figure 3.* The variation in the ratio $r_c$ during training. Each curve represents a label class. The ratio $r_c$ for the input features is shown at epoch 0. We see that the ratio generally increases at the beginning of the training and stabilizes. This suggests that the encoders indeed have good clustering capabilities (as compared with the input).

### 4.3. The FD-GCL Model

We are now in a position to provide the full details of the proposed FD-GCL model. Recall that we have a graph $\mathcal{G}$ (with adjacency matrix $\mathbf{A}$) and initial node features $\mathbf{X}$, col-

lectively denoted by $\mathcal{X} = (\mathbf{A}, \mathbf{X})$. Our goal is to learn two FDE encoders, $f_{\theta_1}$ and $f_{\theta_2}$, in an unsupervised manner. We want to have the following characterization of the views they generate: $f_{\theta_1} \longleftrightarrow$ "local", $f_{\theta_2} \longleftrightarrow$ "global".

The pipelines for $f_{\theta_1}$ and $f_{\theta_2}$ are identical. For convenience, we use $l$ to denote the encoder index, which can be either 1 or 2. The procedure consists of the following steps (see Fig. 4 for an illustration):

S1  Apply a *linear encoder* with a learnable matrix $\mathbf{W}_l$ to $\mathbf{X}$ to obtain $\mathbf{Y}_l = \mathbf{X}\mathbf{W}_l$. This step typically increases the feature dimension.

S2  Choose an FDE order parameter $\alpha_l$ and set $\mathbf{Z}_{\alpha_l}(0) = \mathbf{Y}_l$. Use the FDE-based encoders with skip connections to obtain $\mathbf{Z}_{\alpha_l}(t)$ as described in Section 4.1.

S3  Stop the diffusion at a chosen time $T$. Apply a nonlinear activation function $\sigma$ (e.g., ReLU) to $\mathbf{Z}_{\alpha_l}(T)$ to obtain the final output $\mathbf{Z}_l$.

The model parameter $\theta_l$ includes the learnable weight matrix $\mathbf{W}_l$ and the hyperparameters $\alpha_l$ and $T$. Details on how these hyperparameters are chosen are provided in Appendix F.3.

To learn the model parameters, we apply a *contrastive loss* $\mathcal{L}_0$ to the final features $\mathbf{Z}_1$ and $\mathbf{Z}_2$. In this paper, we choose $\mathcal{L}_0$ to be the mean cosine similarity for its simplicity, which we call *cosmean* (see (13) in Appendix F.4).

Based on the discussion in Section 4.2, we modify $\mathcal{L}_0$ to avoid issues such as the collapse of the two views to a single representation. By Fig. 2, we observe that both $\mathbf{Z}_{\alpha_1}(T)$ and $\mathbf{Z}_{\alpha_2}(T)$ have pronounced main components. The unit directional vectors of their respective dominant components are denoted by $\mathbf{c}_1$ and $\mathbf{c}_2$. To prevent the aforementioned collapse, it suffices to penalize the angle between $\mathbf{c}_1$ and $\mathbf{c}_2$ from being too small. Therefore, our modified loss, named *regularized cosmean*, takes the form:

$$\mathcal{L}(\mathbf{Z}_1, \mathbf{Z}_2) = \mathcal{L}_0(\mathbf{Z}_1, \mathbf{Z}_2) + \eta|\langle \mathbf{c}_1, \mathbf{c}_2 \rangle|,$$

where $\eta$ is a regularization weight. The added regularization has the effect of driving the two feature representations apart and our approach does not need any *negative samples*. The contribution of the regularization term is further analyzed in Section 5.3.

Finally, for downstream tasks, it suffices to take a weighted average $\beta\mathbf{Z}_1 + (1 - \beta)\mathbf{Z}_2$. We either choose $\beta = 0.5$ or tune it based on validation accuracy.

## 5. Experiments

### 5.1. Experimental Setup

**Datasets and splits**   We conduct experiments on both homophilic and heterophilic datasets. The heterophilic

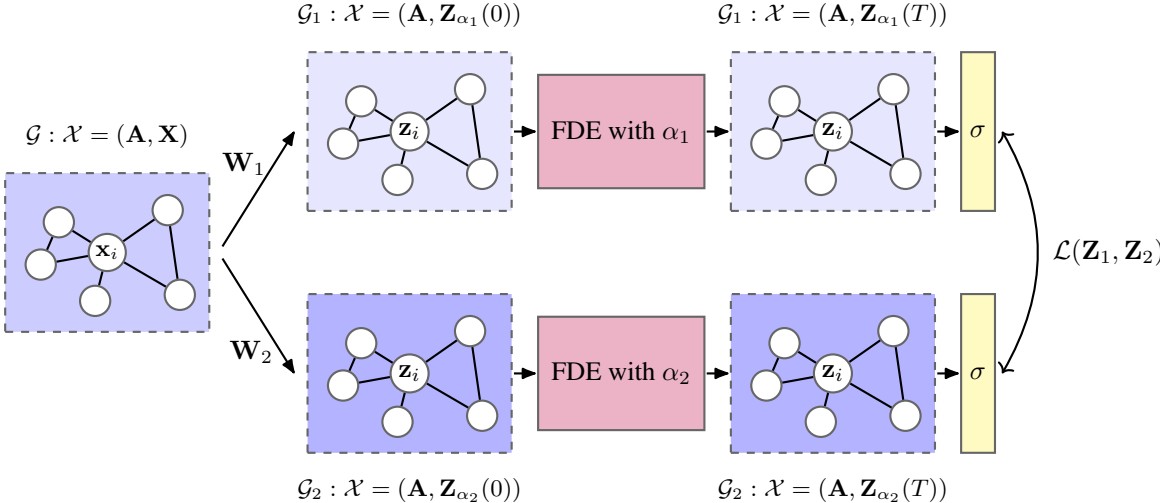

*Figure 4.* The overall proposed contrastive learning framework of FD-GCL. We choose $\alpha_1 < \alpha_2$ for the encoders.

datasets include Wisconsin, Cornell, Texas, Actor, Squirrel, Crocodile, and Chameleon. Additionally, we experiment on two recently proposed large heterophilic datasets: Roman-empire (Roman) and arXiv-year. For homophilic datasets, we use three citation graphs: Cora, Citeseer, and Pubmed, as well as two co-purchase datasets: Computer and Photo. We also include a large-scale homophilic dataset, Ogbn-Arxiv (Arxiv). For all datasets, we use the public and standard splits as used in the cited papers. Detailed descriptions and splits of the datasets are provided in Appendix F.1.

**Baselines** We compare FD-GCL with several recent state-of-the-art unsupervised learning methods: DGI (Velickovic et al., 2019), GMI (Peng et al., 2020), MVGRL (Hassani & Khasahmadi, 2020b), GRACE (Zhu et al., 2020), GCA (Zhu et al., 2021), CCA-SSG (Zhang et al., 2021), BGRL (Thakoor et al., 2022), AFGRL (Lee et al., 2022), Local (L)-GCL (Zhang et al., 2022), HGRL (Chen et al., 2022), DSSL (Xiao et al., 2022), SP-GCL (Wang et al., 2023), GraphACL (Xiao et al., 2023), and PolyGCL (Chen et al., 2024). The detailed descriptions and implementations of these baselines are given in Appendix F.2.

**Evaluation protocol** To evaluate the quality of the representation, we focus on the node classification task. Following the standard linear evaluation protocol, we train a linear classifier on the frozen representations and report the test accuracy as the evaluation metric. Results for graph classification are reported in Appendix G.3.

**Setup** For FD-GCL here, we apply the basic GRAND version of $\mathcal{F}$ in (2), i.e., $-(\mathbf{I} - \overline{\mathbf{A}})\mathbf{Z}(t)$. Other options for $\mathcal{F}$ are explored and evaluated in Appendix C and Appendix G.2.

We initialize model parameters randomly and train the en-coder using the Adam optimizer. Each experiment is conducted with ten random seeds, and we report the average performance along with the standard deviation. To ensure a fair comparison, the best hyperparameter configurations for all methods are selected based solely on validation set accuracy. For baselines lacking results on certain datasets or not utilizing standard public data splits (Chen et al., 2022; 2024; Xiao et al., 2022), we reproduce their outcomes with the official code of the authors. Additional implementation details and the hyperparameter search space are provided in Appendix F.3.

### 5.2. Overall Performance Comparison

We show the node classification results for homophilic and heterophilic datasets in Table 1 and Table 2, respectively. Notably, FD-GCL achieves the best performance across most datasets, excelling in both homophilic and heterophilic datasets. Specifically, FD-GCL demonstrates significant relative improvements on heterophilic datasets compared to the second-best method, achieving improvements of 9.87% on Squirrel, 5.2% on Chameleon, 2.16% on Wisconsin, 8.24% on Cornell, 6.22% on Texas, and 1.33% on Actor. This is while maintaining competitive performance on homophilic graphs. Contrastive strategies such as CCA-SSG, GCA, and BGRL, which rely on augmentations, struggle on heterophilic graphs due to their implicit reliance on the homophily assumption. Meanwhile, our augmentation-free FD-GCL consistently outperforms other augmentation-free methods (e.g., L-GCL, SP-GCL, DSSL, and GraphACL) across both homophilic and heterophilic datasets. This improvement can be attributed to the FDE-based encoders, which enable more effective representations for diverse graph structures. The corresponding values of $\alpha_1$ and $\alpha_2$ for

*Table 1.* Node classification results(%) on homophilic datasets. The best and the second-best result under each dataset are highlighted in red and blue, respectively. OOM refers to out of memory on an NVIDIA RTX A5000 GPU (24GB).

| Method | DGI | GMI | MVGRL | GRACE | GCA | CCA-SSG | BGRL | AFGRL | SP-GCL | GraphACL | PolyGCL | FD-GCL |
|---|---|---|---|---|---|---|---|---|---|---|---|---|
| Cora | 82.30±0.60 | 82.70±0.20 | 82.90±0.71 | 80.00±0.41 | 82.93±0.42 | 84.00±0.40 | 82.70±0.60 | 82.31±0.42 | 83.16±0.13 | 84.20±0.31 | 81.97±0.19 | 84.50±0.43 |
| Citeseer | 71.80±0.70 | 73.01±0.30 | 72.61±0.70 | 71.72±0.62 | 72.19±0.31 | 73.10±0.30 | 71.10±0.80 | 68.70±0.30 | 71.96±0.42 | 73.63±0.22 | 71.97±0.29 | 73.72±0.30 |
| Pubmed | 76.80±0.60 | 80.11±0.22 | 79.41±0.31 | 79.51±1.10 | 80.79±0.45 | 81.00±0.40 | 79.60±0.50 | 79.71±0.21 | 79.16±0.84 | 82.02±0.15 | 77.48±0.39 | 82.02±0.28 |
| Computer | 83.95±0.47 | 82.21±0.34 | 87.52±0.11 | 86.51±0.32 | 87.85±0.31 | 88.74±0.28 | 89.90±0.31 | 89.68±0.19 | 89.80±0.25 | 86.73±0.34 | 90.13±0.45 | |
| Photo | 91.61±0.22 | 90.72±0.21 | 91.72±0.10 | 92.5±0.22 | 91.70±0.10 | 93.14±0.14 | 92.90±0.30 | 93.25±0.33 | 92.49±0.31 | 93.31±0.19 | 91.67±0.21 | 94.17±0.86 |
| Arxiv | 70.32±0.25 | OOM | OOM | OOM | 69.37±0.20 | 71.21±0.20 | 71.64±0.24 | OOM | 68.25±0.22 | 71.72±0.26 | OOM | 70.46±0.13 |

*Table 2.* Node classification results(%) on heterophilic datasets. The best and the second-best result under each dataset are highlighted in red and blue, respectively.

| Method | DGI | GCA | CCA-SSG | BGRL | HGRL | L-GCL | DSSL | SP-GCL | GraphACL | PolyGCL | FD-GCL |
|---|---|---|---|---|---|---|---|---|---|---|---|
| Squirrel | 26.44±1.12 | 48.09±0.21 | 46.76±0.36 | 36.22±1.97 | 48.31±0.65 | 52.94±0.88 | 40.51±0.38 | 52.10±0.67 | 54.05±0.13 | 34.25±0.66 | 64.92±1.46 |
| Chameleon | 60.27±0.70 | 63.66±0.32 | 62.41±0.22 | 64.86±0.63 | 65.82±0.61 | 68.74±0.49 | 66.15±0.32 | 65.28±0.53 | 69.12±0.24 | 46.84±1.53 | 74.32±1.24 |
| Crocodile | 51.25±0.51 | 60.73±0.28 | 56.77±0.39 | 53.87±0.65 | 61.87±0.45 | 60.18±0.43 | 62.98±0.51 | 61.72±0.21 | 66.17±0.24 | 65.95±0.59 | 68.73±0.77 |
| Actor | 28.30±0.76 | 28.47±0.29 | 27.82±0.60 | 28.80±0.54 | 27.95±0.30 | 32.55±1.18 | 28.15±0.31 | 28.94±0.69 | 30.03±0.13 | 34.37±0.69 | 35.70±1.08 |
| Wisconsin | 55.21±1.02 | 59.55±0.81 | 58.46±0.96 | 51.23±1.17 | 63.90±0.58 | 65.28±0.52 | 62.25±0.55 | 60.12±0.39 | 69.22±0.40 | 76.08±3.33 | 79.22±5.13 |
| Cornell | 45.33±6.11 | 52.31±1.09 | 52.17±1.04 | 50.33±2.29 | 51.78±1.03 | 52.11±2.37 | 53.15±1.28 | 52.29±1.21 | 59.33±1.48 | 43.78±3.51 | 67.57±5.27 |
| Texas | 58.53±2.98 | 52.92±0.46 | 59.89±0.78 | 52.77±1.98 | 61.83±0.71 | 60.68±1.18 | 62.11±1.53 | 59.81±1.33 | 71.08±0.34 | 72.16±3.51 | 78.38±5.80 |
| Roman | 63.71±0.63 | 65.79±0.75 | 67.35±0.61 | 68.66±0.39 | 71.84±0.41 | 69.74±0.53 | 71.70±0.54 | 70.88±0.35 | 74.91±0.28 | 72.97±0.25 | 72.56±0.63 |
| Arxiv-year | 39.26±0.72 | 42.96±0.39 | 37.38±0.41 | 43.02±0.62 | 43.71±0.54 | 43.92±0.52 | 45.80±0.57 | 44.11±0.35 | 47.21±0.39 | 43.07±0.23 | 47.22±0.13 |

each dataset are reported in Table 8 in Appendix F.3.

## 5.3. Ablation Studies

**The effect of order parameters** We evaluate the performance of the FD-GCL model under various configurations of the parameters $\alpha_1$ and $\alpha_2$. Specifically, we consider cases where $\alpha_1$ and $\alpha_2$ (i) differ significantly, (ii) differ slightly, and (iii) are equal. The configurations include $\alpha_1 = \alpha_2 = 1$, $\alpha_1 = \alpha_2 = 0.5$, and $\alpha_1 = \alpha_2 = 0.1$. Additionally, we compare these configurations with a standard 2-layer GCN with skip connections as the encoder. The experiments are conducted on the Cora (homophilic), Wisconsin (heterophilic), and Crocodile (heterophilic) datasets, using classification accuracy as the evaluation metric. The results are shown in Table 3. The findings suggest that configurations with distinct $\alpha_1$ and $\alpha_2$ generally outperform those with equal values. Larger $\alpha_2 - \alpha_1$ encourages the FDE-based encoder to generate more diverse graph views, enhancing the contrastive learning process and enabling the model to capture richer and more discriminative representations. In contrast, the equality of parameters $\alpha_1 = \alpha_2$ may lead to less diverse views, potentially limiting the model's ability to learn comprehensive representations.

**Loss functions** To evaluate the performance of FD-GCL under different contrastive learning paradigms, we compare several widely used *contrastive loss* functions in terms of their impact on the final feature representations $\mathbf{Z}_1$ and $\mathbf{Z}_2$. Specifically, we assess classification accuracy on two benchmark datasets: Cora (homophilic) and Wisconsin (heterophilic). The contrastive loss functions considered include Euclidean loss, Cosmean, Barlow Twins loss (Zbontar et al.,

*Table 3.* Node classification results (%) across different datasets and parameter configurations.

| | Method | Cora | Wisconsin | Crocodile |
|---|---|---|---|---|
| | GCN | 56.23±0.54 | 65.10±5.60 | 62.58±0.85 |
| FD-GCL | $\alpha_1 = \alpha_2 = 1$ | 78.09±0.19 | 61.57±6.21 | 63.57±1.01 |
| | $\alpha_1 = \alpha_2 = 0.5$ | 81.19±0.12 | 66.27±3.59 | 60.32±0.92 |
| | $\alpha_1 = \alpha_2 = 0.1$ | 77.52±0.13 | 77.06±4.64 | 67.89±0.98 |
| | $\alpha_1 = 0.1, \alpha_2 = 0.2$ | 78.65±0.17 | 74.71±3.77 | 68.06±1.16 |
| | $\alpha_1 = 0.5, \alpha_2 = 1$ | 82.53±0.13 | 63.33±6.27 | 68.42±0.71 |
| | $\alpha_1 = 0.01, \alpha_2 = 1$ | **84.27±0.27** | **79.22±5.13** | **68.99±0.66** |

2021), VICReg loss (Bardes et al., 2022), and our proposed Regularized Cosmean loss. The results, summarized in Fig. 5, highlight the effectiveness of the Regularized Cosmean loss. Unlike other loss functions, which exhibit performance degradation as training progresses, the Regularized Cosmean loss maintains consistent accuracy across training epochs, demonstrating superior stability. This consistency can be attributed to its ability to mitigate dimension collapse, ensuring reliable performance over extended training periods. Additional details on the definitions of these loss functions and further comparison results on other datasets are provided in Appendix F.4.

## 5.4. Complexity Analysis

The training time complexity of FD-GCL consists of two components: the learning of FDE encoders and the loss computation. Suppose the graph consists of $N$ nodes and $|\mathcal{E}|$ edges. The numerical solution of FDE is computed iteratively for $E := T/h$ time steps, where $h$ represents the discretization size and $T$ the integration time. At each step, the function $\mathcal{F}(\mathbf{W}, \mathbf{Z}_j)$ is evaluated, with intermediate re-

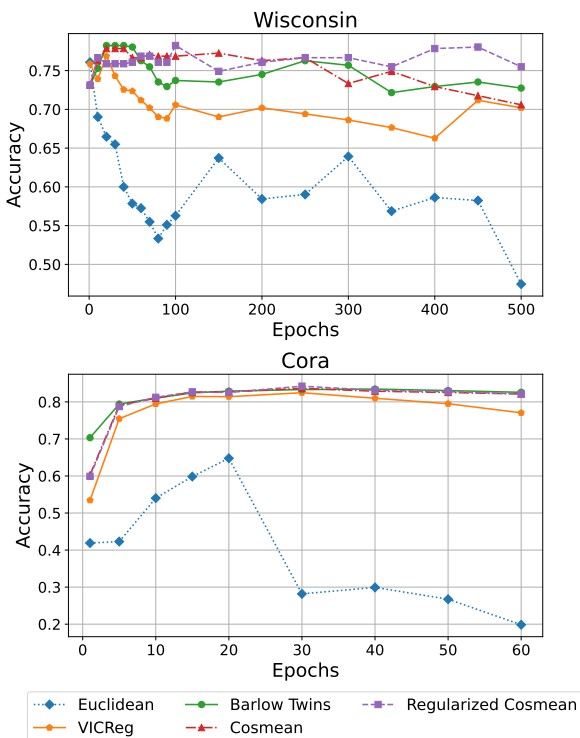

Figure 5. Accuracy vs. training epochs for various loss functions on the Cora and Wisconsin datasets.

sults $\{\mathcal{F}(\mathbf{W}, \mathbf{Z}_j)\}_j$ stored for efficiency. Assuming $\mathcal{F}$ corresponds to the GRAND model (Chamberlain et al., 2021b), the cost per evaluation is $C = |\mathcal{E}|d$, where $|\mathcal{E}|$ represents the edge set size and $d$ the dimensionality of the features. The total complexity for solving the FDE is $O\left(E|\mathcal{E}|d + E^2\right)$. With fast convolution algorithms (Mathieu et al., 2013), this can be reduced to $O(EC + E \log E)$. Meanwhile, the computation of $\mathcal{L}$ costs $O(N)$ time. Thus the overall training time complexity of FD-GCL is $O(EC + E \log E + N)$.

Comparisons of training time and storage with baselines are summarized in Table 4 and Table 5, respectively. The results, particularly training time comparisons, support that our model is simple and efficient. It scales well as the graph size increases.

Table 4. Training time (s) across different datasets with single training epoch. OOM refers to out of memory on an NVIDIA RTX A5000 GPU (24GB).

| Method | Cora | Wisconsin | Arxiv |
|---|---|---|---|
| GraphACL | 0.22 | 0.63 | 927.48 |
| PolyGCL | 0.34 | 0.23 | OOM |
| FD-GCL | 0.28 | 0.30 | 3.16 |

Table 5. Storage (MiB) across different datasets. OOM refers to out of memory on an NVIDIA RTX A5000 GPU (24GB).

| Method | Cora | Wisconsin | Arxiv |
|---|---|---|---|
| GraphACL | 326 | 204 | 6114 |
| PolyGCL | 4098 | 894 | OOM |
| FD-GCL | 1020 | 556 | 18192 |

## 6. Conclusion

We have proposed a simple and effective augmentation-free graph contrastive learning framework using graph neural diffusion models governed by fractional differential equations. By varying the order parameter, our method generates diverse views that capture both local and global graph information, eliminating the need for both complex augmentations and negative samples. It achieves state-of-the-art performance across diverse datasets. Future work could explore adaptive or data-driven strategies to improve efficiency and scalability, particularly, in tuning the order parameters.

## 7. Limitations

While our framework achieves state-of-the-art performance and removes the need for complex augmentations and negative samples, it has some limitations. The order parameters must be tuned manually, which may impede large-scale or real-time applications. The diffusion process can be expensive on massive graphs, and generalizability to highly irregular or evolving topologies remains understudied. Furthermore, performance may degrade in low-data settings, where there is insufficient data to guide parameter optimization. Future work could address adaptive parameter selection, scalability improvements, and robust methods for highly dynamic graphs.

## Impact Statement

This paper introduces a pioneering framework for augmentation-free contrastive learning by means of using FDE-based graph neural diffusion models, poised to significantly influence both the development of GNNs and their applications across diverse domains. By leveraging continuous dynamics governed by fractional differential equations, our approach enhances the flexibility and robustness of representation learning while eliminating the need for both complex augmentations and negative samples. The societal impact of this work depends on a commitment to ethical standards and responsible use, ensuring that advancements in contrastive learning lead to positive outcomes without exacerbating biases, inequality, or misuse in sensitive applications.

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

# A. List of Notations

For easy reference, we list the most used notations in Table 6.

*Table 6.* List of notations

| | |
|---|---|
| Graph | $\mathcal{G} = (\mathcal{V}, \mathcal{E})$ |
| Nodes | $v_1, \ldots, v_N$ |
| Adjacency (and normalized adj.) matrix | $\mathbf{A} = (a_{ij}), \overline{\mathbf{A}}$ |
| Normalized Laplacian | $\overline{\mathbf{L}} = \mathbf{U}\mathbf{\Lambda}\mathbf{U}^\mathsf{T}$ |
| Eigenvectors and eigenvalues | $\mathbf{u}_i, \lambda_i$ |
| Encoders | $f_\theta, f_{\theta_1}, f_{\theta_2}$ |
| Differential order parameters | $\alpha, \alpha_1, \alpha_2$ |
| Differential operators | $\mathrm{d}/\mathrm{d}t, D_t^\alpha$ |
| Initial features | $\mathbf{X} = [\mathbf{x}_i^\mathsf{T}]$ |
| Learned features | $\mathbf{Z}, \mathbf{Z}_1, \mathbf{Z}_2, \mathbf{Z}_{\alpha_1}(t), \mathbf{Z}_{\alpha_2}(t)$ |
| Linear encoder parameters | $\mathbf{W}_l$ |
| Loss functions | $\mathcal{L}_0, \mathcal{L}$ |

# B. More on Related Works

## B.1. Graph Contrastive Learning without Augmentation

Deep Graph Infomax (DGI) (Velickovic et al., 2019) is a foundational framework in graph contrastive learning, maximizing mutual information (MI) between local node features and a global graph representation. It employs a readout function to aggregate node features into a global representation and a discriminator to distinguish positive samples, derived from the original graph, from negative samples, corrupted graph via shuffled node features. This corruption serves as augmentation, boosting robustness and generalizability. Contrastive Multi-view Representation Learning (MVGRL) (Hassani & Khasahmadi, 2020a) extends this approach by incorporating multiple graph views derived from different graph diffusion processes. Its discriminator contrasts node-level and graph-level representations across views, enhancing representation quality. Cross-Scale Contrastive Graph Knowledge Synergy (CGKS) (Zhang et al., 2023) further builds a graph pyramid of coarse-grained views and introduces a joint optimization strategy with a pairwise contrastive loss to promote knowledge transfer across scales.

GRACE (Zhu et al., 2020) adopts a unique strategy by generating two graph views through edge removal and node feature masking, then maximizing the agreement between their node embeddings. It also leverages both inter-view and intra-view negative pairs to enrich the learning process. GCA (Zhu et al., 2021) builds on this by introducing adaptive augmentation for graph-structured data, leveraging priors from both topological and semantic graph information. Unlike prior methods that rely on two correlated views, ASP (Chen & Kou, 2023) incorporates three distinct views—original, attribute, and global—into a joint contrastive learning framework, enhancing representation learning across these perspectives.

GraphCL (You et al., 2020) introduces various augmentation strategies specifically designed for graph data. DSSL (Xiao et al., 2022) and HGRL (Chen et al., 2022) extend unsupervised learning to nonhomophilous graphs by capturing global and high-order information. HGRL relies on graph augmentations, while DSSL assumes a graph generation process, which may not always reflect real-world graphs. While these methods have advanced graph contrastive learning, augmentation-based approaches face limitations. Their performance is sensitive to the choice of augmentation, and no universally optimal strategy exists. Additionally, augmentation-based GCL methods tends to focus the encoder on capturing low-frequency information while neglecting high-frequency components, which reduces performance on heterophilic graphs (Liu et al., 2022).

To address these limitations, augmentation-free methods have emerged. Followed by DGI, Graphical Mutual Information (GMI) (Peng et al., 2020) directly measures mutual information (MI) between input data and representations of nodes and

edges without relying on data augmentation, providing a more direct approach to optimizing information preservation. Similarly, L-GCL (Zhang et al., 2022) avoids augmentations but focuses on homophilous graphs. SP-GCL (Wang et al., 2023), on the other hand, effectively handles heterophilic graphs by capturing both low- and high-frequency components. GraphACL (Xiao et al., 2023) further eliminates both augmentations and homophily assumptions, ensuring robust and consistent performance across diverse graph structures. PolyGCL (Chen et al., 2024) leverages polynomial filters with learnable parameters to generate low-pass and high-pass spectral views, achieving contrastive learning without relying on complex data augmentations.

### B.2. Graph Contrastive Learning without Negative Sample Pairs

Building on the success of BYOL in image data, BGRL (Thakoor et al., 2022) eliminates the need for negative samples in graph contrastive learning. It generates two graph augmentations using random node feature masking and edge masking and employs an online encoder and a target encoder. The objective is to maximize the cosine similarity between the online encoder's prediction and the target encoder's embedding. A stop-gradient operation in the target encoder prevents mode collapse, ensuring stable training.

Augmentation-Free Graph Representation Learning (AFGRL) (Lee et al., 2022) addresses the limitations of augmentation-dependent methods like BGRL and GCA (Zhu et al., 2021), where representation quality heavily depends on the choice of augmentation schemes. Building on the BGRL framework, AFGRL eliminates the need for augmentations by generating positive samples directly from the original graph for each node. This approach captures both local structural information and global semantics. However, it introduces higher computational costs.

Inspired by Canonical Correlation Analysis (CCA) methods (Hardoon et al., 2004), CCA-SSG (Zhang et al., 2021) introduces an unsupervised learning framework for graphs without relying on negative sample pairs. It maximizes the correlation between two augmented views of the same input while decorrelating the feature dimensions within a single view's representation.

These advancements highlight promising alternatives to traditional graph contrastive learning methods. Employing augmentation-free frameworks or innovative masking strategies mitigates challenges associated with negative sample selection and augmentation dependency, offering robust solutions for graph representation learning.

## C. Integer-order Continuous GNNs

Recent works, including GRAND (Chamberlain et al., 2021b), GraphCON (Rusch et al., 2022), GREAD (Choi et al., 2023), CDE (Zhao et al., 2023), have employed ordinary or partial differential equations (ODEs/PDEs) on graphs for feature aggregation. These continuous GNN models typically use the usual integer-order derivatives. We shall highlight the governing differential equation in each of these models.

**GRAND:** GRAND models heat diffusion on graphs. For the basic GRAND-l model, the governing differential equation for the diffusion process is:

$$\frac{\mathrm{d}\mathbf{Z}(t)}{\mathrm{d}t} = -\overline{\mathbf{L}}\mathbf{Z}(t). \tag{3}$$

More generally, the adjacency matrix can be updated during learning via the attention between node features, and the resulting model is called GRAND-nl.

**GraphCON**: GraphCON is governed by a second-order ODE modeling oscillator dynamical systems:

$$\frac{\mathrm{d}^2\mathbf{Z}(t)}{\mathrm{d}t^2} = \sigma(\mathbf{F}_\theta(\mathbf{Z}(t), t)) - \gamma\mathbf{Z}(t) - \nu\frac{\mathrm{d}\mathbf{Z}(t)}{\mathrm{d}t}, \tag{4}$$

where $\mathbf{F}_\theta(\cdot)$ represents a learnable 1-neighborhood coupling function, $\sigma$ is an activation function, and $\gamma$ and $\nu$ are adjustable parameters.

**CDE**: CDE is Based on the convection-diffusion equation. It includes a diffusion term and a convection term, and the latter is to address information propagation from heterophilic neighbors. More specifically, the ODE takes the following form:

$$\frac{\mathrm{d}\mathbf{Z}(t)}{\mathrm{d}t} = -\overline{\mathbf{L}}\mathbf{Z}(t) + \mathrm{div}(\mathbf{V}(t) \circ \mathbf{Z}(t)). \tag{5}$$

Compared with GRAND, there is an addition of the term involving $\mathbf{V}$. To explain, $\mathbf{V}_{ij}(t) \in \mathbb{R}^d$ is the velocity vector associated with each edge $(i, j)$ at time $t$. One has ($\mathcal{E}$ be the edge set):

$$i\text{-th row of } (\text{div}(\mathbf{V}(t) \circ \mathbf{Z}(t))) = \sum_{j:(i,j)\in\mathcal{E}} \mathbf{V}_{ij}(t) \odot \mathbf{z}_j(t). \tag{6}$$

The velocity $\mathbf{V}_{ij}(t)$ is given by

$$\mathbf{V}_{ij}(t) = \sigma\left(\mathbf{M}(\mathbf{z}_j(t) - \mathbf{z}_i(t))\right), \tag{7}$$

with $\mathbf{M}$ is a learnable matrix and $\sigma$ is an activation function.

**GREAD:** GREAD is also proposed to handle heterophilic graphs. It has a similar high-level idea to CDE by adding a reaction term to the ODE of GRAND, thereby establishing a diffusion-reaction equation for GNNs. The governing equation for this model is expressed as:

$$\frac{\mathrm{d}\mathbf{Z}(t)}{\mathrm{d}t} = -\gamma\overline{\mathbf{L}}(\mathbf{Z}(t)) + \nu r(\mathbf{Z}(t)), \tag{8}$$

where $r(\mathbf{Z}(t))$ is the reaction term, $\gamma$ and $\nu$ are trainable weight parameters.

## D. Fractional-order Derivatives and GNNs

Kang et al. (2024); Zhao et al. (2024) extend the ODE-based approaches by incorporating graph neural Fractional-order Differential Equations (FDEs), generalizing the order of the derivative to positive real number $\alpha$. The motivation is the solution to an FDE that encodes historical information and thus retains the "memory" of the evolution.

### D.1. Fractional-order Derivatives

Recall that the first-order derivative of a scalar function $f(t)$ is defined as the rate of change of $f$ at a time $t$:

$$f'(t) = \frac{\mathrm{d}f(t)}{\mathrm{d}t} = \lim_{\Delta t \to 0} \frac{f(t + \Delta t) - f(t)}{\Delta t}$$

It is local in the sense that $f'(t)$ is determined by the value of $f$ in a small neighborhood of $t$.

Fractional-order derivatives $D_t^\alpha$ generalize integer derivatives and the domain of $\alpha$ is extended to any positive real number. By the relation $D_t^{\alpha+1} = D_t^\alpha D_t^1$, it suffices to define $D_t^\alpha$ for the order parameter $\alpha \in (0, 1)$. For the encoder design in this paper, we only consider $\alpha \in (0, 1]$. The domain is large enough for us to choose encoders that generate different views. Moreover, solving a higher-order FDE requires additional information such as the initial derivative, which is usually not available.

Although there are different approaches to defining $D_t^\alpha$, they all share the common characterization that the derivative is defined using an integral. Hence, the operator is "global" as historical values of the function are used. We show two approaches below.

The *left fractional derivative* (Stinga, 2023) of $f(t)$ is defined by the following integral

$$D_t^\alpha f(t) = \frac{1}{\Gamma(-\alpha)} \int_0^\infty \frac{f(t-\tau) - f(t)}{\tau^{1+\alpha}} d\tau,$$

where $\Gamma(\cdot)$ is the Gamma function (see (9) below).

On the other hand, the *Caputo fractional derivative* (Diethelm, 2010) is defined as

$$D_t^\alpha f(t) = \frac{1}{\Gamma(1-\alpha)} \int_0^t \frac{f'(\tau)}{(t-\tau)^\alpha} \, \mathrm{d}\tau.$$

One can find other approaches and more discussions in Tarasov (2011).

### D.2. Incorporating FDE

To build a diffusion model based on FDE, it usually suffices to replace the integer order derivative with a fractional-order derivative in the governing differential equation. For example, the ODE in the GraphCON model (Rusch et al., 2022) is equivalent to a system of equations:

$$\frac{\mathrm{d}\mathbf{Y}(t)}{\mathrm{d}t} = \sigma\left(\mathbf{F}_\theta(\mathbf{Z}(t), t)\right) - \gamma\mathbf{Z}(t) - \nu\mathbf{Y}(t)$$
$$\frac{\mathrm{d}\mathbf{Z}(t)}{\mathrm{d}t} = \mathbf{Y}(t).$$

Replacing $\mathrm{d}/\mathrm{d}t$ with $D_t^\alpha$, one obtains its FDE version as

$$D_t^\alpha\mathbf{Y}(t) = \sigma\left(\mathbf{F}_\theta(\mathbf{Z}(t), t)\right) - \gamma\mathbf{Z}(t) - \nu\mathbf{Y}(t)$$
$$D_t^\alpha\mathbf{Z}(t) = \mathbf{Y}(t).$$

A numerical solver is provided in Kang et al. (2024).

## E. Theoretical Discussions

In this section, we provide a rigorous discussion of the formal version of Theorem 1. To do this, we need the Mittag-Leffler functions, which were introduced by Mittag-Leffler in 1903 in the form of a Maclaurin series. We focus on a special case defined as follows:

$$e_\alpha(\lambda, t) = \sum_{n=0}^{\infty}(-1)^n\frac{\lambda^n t^{\alpha n}}{\Gamma(\alpha n + 1)}, \quad \lambda > 0, \ t \geq 0.$$

Here, $\Gamma(\cdot)$ denotes the Gamma function, which is a meromorphic function on $\mathbb{C}$ defined by the integral

$$\Gamma(z) = \int_0^\infty t^{z-1}e^{-t}\,\mathrm{d}t, \tag{9}$$

for $\Re(z) > 0$.

We make the following *assumptions* regarding the encoder. The function $\mathcal{F}$ in (2) is the simple $-\overline{\mathbf{L}}\mathbf{X}$ as described after (1). For $D_t^\alpha$, we consider the left fractional derivative. For the model, we apply skip-connection after a diffusion of length $\tau$, and there are $m$-iterations so that the total time is $T = m\tau$. We shall consider $\alpha \in (0, 1)$, and this a mild assumption, as we have seen that $D_t^1 = \mathrm{d}/\mathrm{d}t$ is the limit of $D_t^\alpha$ when $\alpha \to 1$. The following formal version of Theorem 1 analyzes the (asymptotic) properties of the Fourier coefficients of the output features.

**Theorem 2.** *Suppose $\mathcal{G}$ is a connected graph and $0 < \alpha_1 < \alpha_2 < 1$ are the order parameters. For $l = 1, 2$, let $n_l \geq 1$ be the integer such that $n_l\alpha_l < 1 \leq (n_l + 1)\alpha_l$. Consider a graph signal $\mathbf{x}$ with Fourier coefficients $\{c_i, 1 \leq i \leq N\}$. Let $\mathbf{z}_{\alpha_l}(T)$ be the output features of the encoders with order parameters $\alpha_l$. Its Fourier coefficients are $\{c_{\alpha_l, i}(T), 1 \leq i \leq N\}$. Then the following holds:*

*(a) We have the expression:*

$$c_{\alpha_l, i}(T) = \left(\sum_{j=0}^{n_l} b_{\alpha_l, i, j}\tau^{-j\alpha_l} + O(\frac{1}{\tau})\right)c_i,$$

*and $b_{\alpha_l, i, j} > 0$.*

*(b) For fixed $l$ and $j$, the coefficient $b_{\alpha_l, i, j}$ is decreasing w.r.t. $i$.*

*(c) For fixed $i$ and $j$, we have $b_{\alpha_1, i, j} > b_{\alpha_2, i, j}$.*

*Proof.* Let $\overline{\mathbf{L}} = \mathbf{U}\boldsymbol{\Lambda}\mathbf{U}^\mathsf{T}$ be the orthogonal eigendecomposition of $\overline{\mathbf{L}}$. As $\overline{\mathbf{L}}$ is positive semi-definite, its (ordered) eigenvalues satisfy $0 = \lambda_1 < \lambda_2 \leq \lambda_3 \ldots \leq \lambda_N$. Notice we have $\lambda_2 > 0$ as $\mathcal{G}$ is connected. To solve the FDE $D_t^\alpha \mathbf{z}_\alpha(t) = -\overline{\mathbf{L}}\mathbf{z}_\alpha(t)$, we perform eigendecomposition:

$$D_t^\alpha \mathbf{z}_\alpha(t) = -\mathbf{U}\boldsymbol{\Lambda}\mathbf{U}^\mathsf{T}\mathbf{z}_\alpha(t), \text{ and hence } D_t^\alpha \mathbf{U}^\mathsf{T}\mathbf{z}_\alpha(t) = -\boldsymbol{\Lambda}\mathbf{U}^\mathsf{T}\mathbf{z}_\alpha(t).$$

Therefore, the spectral decomposition of the dynamics $\mathbf{z}_\alpha(t)$ follows the equation $D_t^\alpha \mathbf{c}_\alpha(t) = -\boldsymbol{\Lambda}\mathbf{c}_\alpha(t)$, where the $i$-th entry of $\mathbf{c}_\alpha(t)$ is $c_{\alpha,i}(t)$.

For consistency, we set $e_\alpha(0, t) = 1$ the constant function independent of $\alpha$. Therefore, by Stinga (2023), we have the solution $c_{\alpha,i}(t) = e_\alpha(\lambda, t)c_{\alpha,i}(0), 1 \leq i \leq N$, where $c_{\alpha,i}(0) = c_i$ depends only on the input feature. Recall we perform diffusion for a time $\tau$ followed by a skip-connection for $m$ iterations. The coefficients of the spectral decomposition of the final output features are

$$c_{\alpha,i}(T) = \Big(1 + \ldots + e_\alpha(\lambda, \tau)\big(1 + e_\alpha(\lambda, \tau)\big)\Big)c_{\alpha,i}(0)$$
$$= \Big(1 + e_\alpha(\lambda, \tau) + \ldots + e_\alpha(\lambda, \tau)^m\Big)c_{\alpha,i}(0). \tag{10}$$

The case $c_{\alpha,i}(0) = c_i = 0$ is trivial as we can set $b_{\alpha_l,i,j}$ to be any positive number. We assume $c_{\alpha,i}(0) \neq 0$ for the rest of the proof, and want to estimate $1 + e_\alpha(\lambda, \tau) + \ldots + e_\alpha(\lambda, \tau)^m$ for $\alpha = \alpha_1$ and $\alpha_2$.

By Erdelyi et al. (1955), if $\lambda > 0$, $e_{\alpha_l}(\lambda, \tau)$ satisfies the following asymptotic estimation:

$$e_{\alpha_l}(\lambda, \tau) = \sum_{j=1}^{n_l} \frac{1}{\lambda_i^j \Gamma(1 - j\alpha_l)} \tau^{-j\alpha_l} + O\Big(\frac{1}{\tau}\Big). \tag{11}$$

The Gamma function $\Gamma(\cdot)$ is positive and strictly decreasing on the interval $(0, 1]$ (e.g., $\Gamma(1) = 1! = 1$). Therefore, as $1 - j\alpha_l > 0$, the coefficient of $\tau^{-j\alpha_l}$ in (11) is positive. Moreover, for $l = 1, 2$ and $i \geq 2$, we have

$$\frac{1}{\lambda_i^j \Gamma(1 - j\alpha_1)} > \frac{1}{\lambda_i^j \Gamma(1 - j\alpha_2)} \text{ and } \frac{1}{\lambda_i^j \Gamma(1 - j\alpha_l)} \geq \frac{1}{\lambda_{i+1}^j \Gamma(1 - j\alpha_l)}. \tag{12}$$

Now to compare the solution for $l = 1, 2$, we express $e_{\alpha_1}(\lambda, \tau)^k$ as follows

$$e_{\alpha_1}(\lambda, \tau)^k = \Big(\sum_{j=1}^{n_2} \frac{1}{\lambda_i^j \Gamma(1 - j\alpha_1)} \tau^{-j\alpha_1} + \sum_{j=n_2+1}^{n_1} \frac{1}{\lambda_i^j \Gamma(1 - j\alpha_1)} \tau^{-j\alpha_1}\Big)^k + O\Big(\frac{1}{\tau}\Big).$$

As compared with

$$e_{\alpha_2}(\lambda, \tau)^k = \Big(\sum_{j=1}^{n_2} \frac{1}{\lambda_i^j \Gamma(1 - j\alpha_2)} \tau^{-j\alpha_2}\Big)^k + O\Big(\frac{1}{\tau}\Big),$$

we see that up to $O(\frac{1}{\tau})$, $e_{\alpha_2}(\lambda, \tau)^k$ and hence $c_{\alpha_2,i}(T)/c_{\alpha_2,i}(0)$ (cf. (10)) consists of a summation of terms $\tau^{-j\alpha_2}$ for $1 \leq j \leq n_2$, and the positive coefficient $\tau^{-j\alpha_2}$ is smaller than that of $\tau^{-j\alpha_1}$ in $c_{\alpha_1,i}(T)/c_{\alpha_1,i}(0)$. Moreover, $c_{\alpha_1,i}(T)/c_{\alpha_1,i}(0)$ has addition terms $\tau^{-j\alpha_1}, n_2 < j \leq n_1$ with positive coefficients. This proves (a) and (c). From the second equality in (12), we obtain the conclusion that $b_{\alpha_l,i,j}$ is decreasing w.r.t. $i$, as claimed in (b). $\qquad\square$

To explain Theorem 1, $t$ in Theorem 1 corresponds to $T = m\tau$ in Theorem 2, and once the number of iterations $m$ is fixed, $\tau \to \infty$ if and only if $t \to \infty$. The signal $\mathbf{x}$ in Theorem 2 is a component (i.e., a column) of the feature matrix $\mathbf{X}$. By Theorem 2, the Fourier coefficients $c_{\alpha_l,i}(t)$ of $\mathbf{Z}_{\alpha_l}(t), l = 1, 2$, relative to the Fourier coefficients of the input features, decreases as the frequency index $i$ increases. Moreover, the decrement for $\mathbf{Z}_{\alpha_1}(t)$ is much slower than that of $\mathbf{Z}_{\alpha_2}(t)$ as $t$ increases. The difference in the rate increases accordingly as $\alpha_2 - \alpha_1$ becomes larger. As the leading Fourier coefficients ($i = 0$) are both the same ($= m$), the Fourier coefficients of $\mathbf{Z}_{\alpha_1}(t)$ are more spread, whence the claims of Theorem 1.

For most datasets studied in the paper, we choose $\alpha_1 \leq 0.1$ and $\alpha_2 > 0.5$, and thus $n_2 = 1$. By (10) and (11), the following holds for $c_{\alpha_2,i}(T)$:

$$c_{\alpha_2,i}(T)/c_{\alpha_2,i}(0) = 1 + \frac{1}{\lambda_i \Gamma(1-\alpha_2)} \tau^{-\alpha_2} + O(\frac{1}{\tau}).$$

For $c_{\alpha_1,i}(T)/c_{\alpha_1,i}(0)$, we can identify a summand $1/(\lambda_i\Gamma(1-\alpha_1)) \cdot \tau^{-\alpha_2}$. The ratio between $1/(\lambda_i\Gamma(1-\alpha_1)) \cdot \tau^{-\alpha_1}$ and $1/(\lambda_i\Gamma(1-\alpha_2)) \cdot \tau^{-\alpha_2}$ is at least $\sqrt{\pi}/1.07 \cdot \tau^{0.4}$. This gives us some understanding of how these two sets of Fourier coefficients differ.

## F. Experimental Details

### F.1. Details of Datasets

*Table 7.* Statistics of homophilic and heterophilic graph datasets

| Dataset | Nodes | Edges | Classes | Node Features | Data splits |
|---|---|---|---|---|---|
| Cora | 2708 | 5429 | 7 | 1433 | standard |
| Citeseer | 3327 | 4732 | 6 | 3703 | standard |
| PubMed | 19717 | 88651 | 3 | 500 | standard |
| Computer | 13752 | 574418 | 10 | 767 | 10%/10%/80% |
| Photo | 7650 | 119081 | 8 | 745 | 10%/10%/80% |
| Ogbn-arxiv | 169343 | 1166243 | 40 | 128 | standard |
| Texas | 183 | 309 | 5 | 1793 | 48%/32%/20% |
| Cornel | 183 | 295 | 5 | 1703 | 48%/32%/20% |
| Wisconsin | 251 | 466 | 5 | 1703 | 48%/32%/20% |
| Chameleon | 2277 | 36101 | 5 | 2325 | 48%/32%/20% |
| Squirrel | 5201 | 217073 | 5 | 2089 | 48%/32%/20% |
| Crocodile | 11631 | 360040 | 5 | 2089 | 48%/32%/20% |
| Actor | 7600 | 33391 | 5 | 932 | 48%/32%/20% |
| Roman | 22662 | 32927 | 18 | 300 | 50%/25%/25% |
| Arxiv-year | 169343 | 1166243 | 5 | 128 | 50%/25%/25% |

Detailed descriptions of the datasets are given below:

**Cora, Citeseer, and Pubmed** (Kipf & Welling, 2017). These datasets are among the most widely used benchmarks for node classification. Each dataset represents a citation graph with high homophily, where nodes correspond to documents and edges represent citation relationships. Node class labels reflect the research field, and node features are derived from a bag-of-words representation of the abstracts. The public dataset split is used for evaluation, with 20 nodes per class designated for training, and 500 and 1,000 nodes fixed for validation and testing, respectively.

**Computer and Photo** (McAuley et al., 2015; Thakoor et al., 2022). These datasets are co-purchase graphs from Amazon, where nodes represent products, and edges connect products frequently bought together. Node features are derived from product reviews, while class labels correspond to product categories. Following the experimental setup in Zhang et al. (2022), the nodes are randomly split into training, validation, and testing sets, with proportions of 10%, 10%, and 80%, respectively.

**Ogbn-arxiv (Arxiv)** (Hu et al., 2020). This dataset is a citation network of Computer Science (CS) papers on arXiv. Each node represents a paper, and edges indicate citation relationships. Node features are 128-dimensional vectors obtained by averaging word embeddings from the paper's title and abstract, generated using the skip-gram model on the MAG corpus. Consistent with Hu et al. (2020), the public split is used for this dataset.

**Texas, Wisconsin and Cornell** (Rozemberczki et al., 2021). These datasets are webpage networks collected by Carnegie Mellon University from computer science departments at various universities. In each network, nodes represent web pages, and edges denote hyperlinks between them. Node features are derived from bag-of-words representations of the web pages. The task is to classify nodes into five categories: student, project, course, staff, and faculty.

**Chameleon, Crocodile and Squirrel** (Rozemberczki et al., 2021). These datasets represent Wikipedia networks, with nodes corresponding to web pages and edges denoting hyperlinks between them. Node features are derived from prominent informative nouns on the pages, while node labels reflect the average daily traffic of each web page.

**Actor** (Pei et al., 2020). This dataset is an actor-induced subgraph extracted from the film-director-actor-writer network. Nodes represent actors, and edges indicate their co-occurrence on the same Wikipedia page. Node features are derived from keywords on the actors' Wikipedia pages, while labels categorize the actors into five groups based on the content of their Wikipedia entries.

For **Texas, Wisconsin, Cornell, Chameleon, Crocodile, Squirrel, and Actor** datasets, we utilize the raw data provided by Geom-GCN (Pei et al., 2020) with the standard fixed 10-fold split for our experiments. These datasets are available for download at: `https://github.com/graphdml-uiuc-jlu/geom-gcn`.

**Roman-empire (Roman)** (Platonov et al., 2023) is a heterophilous graph derived from the English Wikipedia article on the Roman Empire. Each node represents a word (possibly non-unique) in the text, with features based on word embeddings. Node classes correspond to syntactic roles, with the 17 most frequent roles as distinct classes, and all others grouped into an 18th class. Following Platonov et al. (2023), we use the fixed 10 random splits with a 50%/25%/25% ratio for training, validation, and testing.

**Arxiv-year** (Lim et al., 2021) is a citation network derived from a subset of the Microsoft Academic Graph, focusing on predicting the publication year of papers. Nodes represent papers, and edges indicate citation relationships. Node features are computed as the average of word embeddings from the titles and abstracts. Following Lim et al. (2021), the dataset is split into training, validation, and testing sets with a 50%/25%/25% ratio.

### F.2. Baselines

**DGI** (Velickovic et al., 2019): Deep Graph InfoMax (DGI) is a unsupervised learning method that maximizes mutual information between node embeddings and a global graph representation. It employs a readout function to generate the graph-level summary and a discriminator to distinguish between positive (original) and negative (shuffled) node-feature samples, enabling effective graph representation learning.

**GMI** (Peng et al., 2020): Graphical Mutual Information (GMI) measures the mutual information between input graphs and hidden representations by capturing correlations in both node features and graph topology. It extends traditional mutual information computation to the graph domain, ensuring comprehensive representation learning.

**MVGRL** (Hassani & Khasahmadi, 2020b): Contrastive Multi-View Representation Learning (MVGRL) leverages multiple graph views generated through graph diffusion processes. It contrasts node-level and graph-level representations across these views using a discriminator, enabling robust multi-view graph representation learning.

**GRACE** (Zhu et al., 2020): Graph contrastive representation learning (GRACE) model generates two correlated graph views by randomly removing edges and masking features. It focuses on contrasting node embeddings across these views using contrastive loss, maximizing their agreement while incorporating inter-view and intra-view negative pairs, without relying on injective readout functions for graph embeddings.

**GCA** (Zhu et al., 2021): Graph Contrastive Learning with Adaptive Augmentation (GCA) enhances graph representation learning by incorporating adaptive augmentation based on rich topological and semantic priors, enabling more effective graph-structured data representation.

**CCA-SSG** (Zhang et al., 2021): Canonical Correlation Analysis inspired Self-Supervised Learning on Graphs (CCA-SSG) is a graph contrastive learning model that enhances node representations by maximizing the correlation between two augmented views of the same graph while reducing correlations across feature dimensions within each view.

**BGRL** (Thakoor et al., 2022): Bootstrapped Graph Latents (BGRL) is a graph representation learning method that predicts alternative augmentations of the input using simple augmentations, eliminating the need for negative examples.

**AFGRL** (Lee et al., 2022): Augmentation-Free Graph Representation Learning (AFGRL) builds on the BGRL framework, avoiding augmentation schemes by generating positive samples directly from the original graph. This approach captures both local structural and global semantic information, offering an alternative to traditional graph contrastive methods, though at the cost of increased computational complexity.

**L-GCL** (Zhang et al., 2022): Localized Graph Contrastive Learning (LOCAL-GCL) is a unsupervised node representation learning method that samples positive samples from first-order neighborhoods and employs a kernelized negative loss to reduce training time.

**HGRL** (Chen et al., 2022): It is an unsupervised representation learning framework for graphs with heterophily that leverages original node features and high-order information. It learns node representations by preserving original features and capturing informative distant neighbors.

**DSSL** (Xiao et al., 2022): Decoupled self-supervised learning (DSSL) is a flexible, encoder-agnostic representation learning framework that decouples diverse neighborhood contexts using latent variable modeling, enabling unsupervised learning without requiring augmentations.

**SP-GCL** (Wang et al., 2023): Single-Pass Graph Contrastive Learning (SP-GCL) is a single-pass graph contrastive learning method that leverages the concentration property of node representations, eliminating the need for graph augmentations.

**GraphACL** (Xiao et al., 2023): Graph Asymmetric Contrastive Learning (GraphACL) is a simple and effective graph contrastive learning approach that captures one-hop neighborhood context and two-hop monophily similarities in an asymmetric learning framework, without relying on graph augmentations or homophily assumptions.

**PolyGCL** (Chen et al., 2024): It is a graph contrastive learning pipeline that leverages polynomial filters with learnable parameters to generate low-pass and high-pass spectral views, achieving contrastive learning without relying on complex data augmentations.

### F.3. Hyperparameter Choices

We conduct our experiments on a machine equipped with an NVIDIA RTX A5000 GPU with 24 GB of memory. A small grid search is performed to identify the optimal hyperparameters. Specifically, we search for $T$ in $\{1, 1.5, 2, 2.5, 3, 5, 6, 7, 8, 9, 10, 20, 30\}$, $h$ in $\{0.1, 0.15, 0.2, 0.25, 0.3, 0.4, 0.5, 1, 1.5, 2, 3, 5, 10\}$, and the hidden dimension $d$ in $\{256, 512, 1024, 2048, 4096\}$.

Both $\alpha_1$ and $\alpha_2$ are initially considered tunable over the range $(0, 1]$. The *general idea* is to start searching with large $\alpha_2 - \alpha_1$ (suggested by Theorem 1). To simplify the search and maintain a consistent global view, we fix $\alpha_2 = 1$ and tune $\alpha_1$ over the range $(0, 1]$ with a step of $0.01$. Additionally, the learning rate is searched over $\{0.005, 0.01, 0.015, 0.02\}$ and weight decay over $\{0.001, 0.0001, 0.0005, 0.00001\}$. Weight $\beta$ (for combining $\mathbf{Z}_1, \mathbf{Z}_2$) is adjusted within the range $[0, 1]$ with step of $0.05$, and and the regularization weight $\eta$ is tuned within $[0, 0.5)$ with step of $0.01$. The optimal configuration of hyperparameters is determined based on the average accuracy on the validation set. A detailed summary of the selected hyperparameters for each dataset is provided in Table 8.

*Table 8.* Details of the hyperparameters tuned by grid search on various datasets

| Datasets | $T$ | $h$ | $d$ | lr | weight decay | epochs | $\alpha_1$ | $\alpha_2$ | $\beta$ | $\eta$ |
|---|---|---|---|---|---|---|---|---|---|---|
| Cora | 20 | 1 | 256 | 0.01 | 0.0005 | 30 | 0.01 | 1 | 0.55 | 0.15 |
| Citeseer | 6 | 0.4 | 2048 | 0.015 | 0.0005 | 15 | 0.08 | 1 | 0.55 | 0.15 |
| Pubmed | 3 | 0.5 | 4096 | 0.02 | 0.0005 | 1 | 0.75 | 1 | 0.85 | 0.2 |
| Computer | 3 | 0.5 | 2048 | 0.005 | 0.0005 | 1 | 0.94 | 1 | 0.96 | 0.15 |
| Photo | 3 | 0.5 | 4096 | 0.005 | 0.0005 | 1 | 0.9 | 1 | 0.9 | 0.04 |
| Arxiv | 30 | 3 | 256 | 0.01 | 0.0005 | 55 | 0.01 | 1 | 0.55 | 0.2 |
| Squirrel | 40 | 5 | 4096 | 0.01 | 0.0005 | 20 | 0.9 | 1 | 0.6 | 0.01 |
| Chameleon | 30 | 5 | 4096 | 0.01 | 0.0005 | 20 | 0.9 | 1 | 0.9 | 0.05 |
| Crocodile | 20 | 2 | 2048 | 0.01 | 0.0005 | 20 | 0.1 | 1 | 0.55 | 0.01 |
| Actor | 1.5 | 0.15 | 2048 | 0.01 | 0.0005 | 5 | 0.01 | 1 | 0.55 | 0.01 |
| Wisconsin | 20 | 2 | 2048 | 0.01 | 0.0005 | 30 | 0.01 | 1 | 0.6 | 0.1 |
| Cornell | 20 | 2 | 2048 | 0.01 | 0.0005 | 30 | 0.01 | 1 | 0.7 | 0.2 |
| Texas | 30 | 10 | 2048 | 0.01 | 0.0005 | 30 | 0.01 | 1 | 0.6 | 0.01 |
| Roman | 2 | 1.5 | 4096 | 0.01 | 0.0005 | 2 | 0.001 | 1 | 0.5 | 0.05 |
| Arxiv-year | 2 | 1 | 512 | 0.01 | 0.0005 | 2 | 0.99 | 1 | 0.15 | 0.01 |

### F.4. Contrastive Loss Functions

In the absence of explicit negative samples, non-contrastive methods focus on maximizing agreement among positive samples. Examples include knowledge-distillation approaches, such as BGRL (Thakoor et al., 2022) (Cosine similarity-based method), and redundancy-reduction methods, including Barlow Twins (Zbontar et al., 2021) and VICReg (Bardes et al., 2022). The definitions of Euclidean loss, Cosmean loss, Barlow Twins loss, and VICReg loss are provided below.

Let $\mathbf{Z}_1$ and $\mathbf{Z}_2$ denote two feature representations fed into a contrastive loss function. Suppose each feature representation consists of $N$ samples, where $\mathbf{Z}_{1,i}$ and $\mathbf{Z}_{2,i}$ represent the feature vectors corresponding to the $i$-th sample in $\mathbf{Z}_1$ and $\mathbf{Z}_2$, respectively.

**Euclidean Loss** measures the squared $\ell_2$-norm between two feature representations $\mathbf{Z}_1$ and $\mathbf{Z}_2$, encouraging them to be as close as possible. It is defined as:

$$\mathcal{L}_{\text{Euclidean}} = \frac{1}{N} \sum_{i=1}^{N} \|\mathbf{Z}_{1,i} - \mathbf{Z}_{2,i}\|_2^2.$$

**Cosmean Loss** measures the cosine similarity between two feature representations $\mathbf{Z}_1$ and $\mathbf{Z}_2$, encouraging their alignment. It is mathematically expressed as

$$\mathcal{L}_{\text{Cosmean}} = 1 - \frac{1}{N} \sum_{i=1}^{N} \frac{\langle \mathbf{Z}_{1,i}, \mathbf{Z}_{2,i} \rangle}{\|\mathbf{Z}_{1,i}\|_2 \|\mathbf{Z}_{2,i}\|_2} \tag{13}$$

where $\langle \mathbf{Z}_{1,i}, \mathbf{Z}_{2,i} \rangle$ is the inner product of these two vectors $\mathbf{Z}_{1,i}$ and $\mathbf{Z}_{2,i}$, and $\|\mathbf{Z}_{1,i}\|_2$ and $\|\mathbf{Z}_{1,i}\|_2$ are their respective $\ell_2$-norms. The cosmean loss is minimized when the feature vectors are perfectly aligned in the same direction, achieving maximum cosine similarity.

**Barlow Twins Loss** (Zbontar et al., 2021) is designed to encourage similarity between two feature representations $\mathbf{Z}_1$ and $\mathbf{Z}_2$, while reducing redundancy across dimensions within each representation. It is defined as:

$$\mathcal{L}_{\text{Barlow Twins}} = \sum_i \left( \left(1 - \mathbf{C}_{ii}\right)^2 \right) + \lambda \sum_i \sum_{i \neq j} \mathbf{C}_{ii}^2$$

where $\mathbf{C} = \frac{\mathbf{Z}_1^\top \mathbf{Z}_2}{N}$ is the cross-correlation matrix of the normalized feature representations $\mathbf{Z}_1$ and $\mathbf{Z}_2$, and $\lambda$ is a trade-off parameter. The first term minimizes the difference between the diagonal elements of $\mathbf{C}$ and 1, ensuring the features from $\mathbf{Z}_1$ and $\mathbf{Z}_2$ are highly correlated. The second term minimizes the off-diagonal elements, promoting decorrelation between features and reducing redundancy. By balancing these two objectives, Barlow Twins Loss enables learning robust and diverse feature representations.

**VICReg Loss** (Variance-Invariance-Covariance Regularization Loss) (Bardes et al., 2022) is designed to align two feature representations, $\mathbf{Z}_1$ and $\mathbf{Z}_2$, while ensuring variance preservation and minimizing redundancy. It consists of three components: variance regularization, invariance loss, and covariance regularization. The loss is expressed as:

$$\mathcal{L}_{\text{VICReg}} = \eta_1 \mathcal{L}_{\text{inv}} + \eta_2 \mathcal{L}_{\text{var}} + \eta_3 \mathcal{L}_{\text{cov}},$$

where $\eta_1$, $\eta_2$ and $\eta_3$ are hyper-parameters controlling the relative contributions of each term. The invariance component $\mathcal{L}_{\text{inv}}$ is computed as the mean-squared Euclidean distance of the corresponding samples from $\mathbf{Z}_1$ and $\mathbf{Z}_2$:

$$\mathcal{L}_{\text{inv}} = \frac{1}{N} \sum_{i=1}^{N} \|\mathbf{Z}_{1,i} - \mathbf{Z}_{2,i}\|_2^2.$$

The variance component $\mathcal{L}_{\text{var}}$ ensures that each feature dimension in $\mathbf{Z}_1$ and $\mathbf{Z}_2$ has sufficient variance to potentially prevent collapse. It is given by

$$\mathcal{L}_{\text{var}} = \frac{1}{d} \sum_{j=1}^{d} \max(0, \varepsilon - \sqrt{\text{Var}(\mathbf{Z}_1[:,j])}) + \max(0, \varepsilon - \sqrt{\text{Var}(\mathbf{Z}_2[:,j])})$$

where $\mathbf{Z}_1[:,j]$ and $\mathbf{Z}_2[:,j]$ are the $j$-th feature columns of $\mathbf{Z}_1$ and $\mathbf{Z}_2$, respectively. $\text{Var}(\mathbf{Z}_1[:,j]) = \frac{1}{N} \sum_{i=1}^{N} (\mathbf{Z}_1[i,j] - \mu_{1,j})^2$ is the variance of the $j$-th feature in $\mathbf{Z}_1$ (with $\mu_{1,j} = \frac{1}{N} \sum_{i=1}^{N} \mathbf{Z}_1[i,j]$), and $\varepsilon$ is a small positive constant to enforce nonzero variance. And the covariance regularization term $\mathcal{L}_{\text{cov}}$ reduces redundancy by decorrelating different feature dimensions within each representation. It is expressed as:

$$\mathcal{L}_{\text{cov}} = \frac{1}{d} \sum_{i \neq j} \left( \text{Cov}(\mathbf{Z}_1)_{j,k}^2 + \text{Cov}(\mathbf{Z}_2)_{j,k}^2 \right)$$

where $\mathrm{Cov}(\mathbf{Z}_l)_{j,k} = \frac{1}{N} \sum_{i=1}^{N} (\mathbf{Z}_{l,i,j} - \mu_{l,j})(\mathbf{Z}_{l,i,k} - \mu_{l,k})$ represents the off-diagonal elements of the covariance matrix for $\mathbf{Z}_l$, with $l = 1, 2$. The hyperparameters $\eta_1$, $\eta_2$, and $\eta_3$ control the relative contributions of variance regularization, invariance loss, and covariance regularization, respectively. By balancing these three terms, VICReg achieves robust feature alignment while maintaining diversity and decorrelation, making it particularly effective for unsupervised learning tasks.

Fig. 6 presents additional classification accuracy results on two benchmark datasets: Cornell and Squirrel, evaluated above various contrastive loss functions. Consistent with the findings in the main text, these results further underscore the effectiveness of our proposed Regularized Cosmean loss. Unlike other loss functions, which tend to experience performance degradation as training progresses, the Regularized Cosmean loss demonstrates superior stability by maintaining consistent accuracy across epochs. These results provide additional evidence of its ability to mitigate dimension collapse and ensure robust performance.

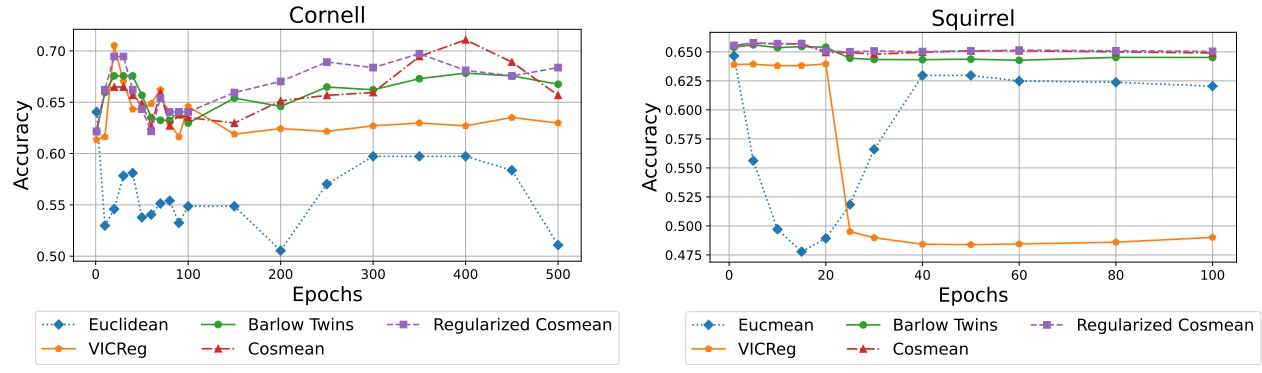

*Figure 6.* Accuracy vs. training epochs for various loss functions on the Cornell and Squirrel datasets

# G. More Numerical Results

## G.1. T-SNE Visualizations of Node Features

In this section, we present additional t-SNE visualizations of node features for each class in the Cora (homophilic) and Wisconsin (heterophilic) datasets. These visualizations are generated using encoders with different FDE order parameters, revealing distinct embedding characteristics produced by the two encoders.

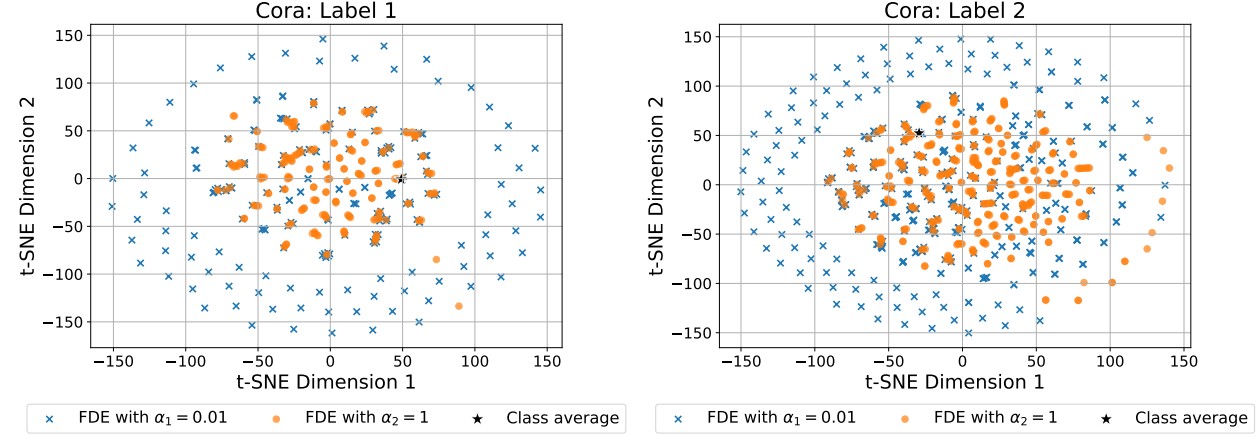

*Figure 7.* Cora: labels 1 and 2

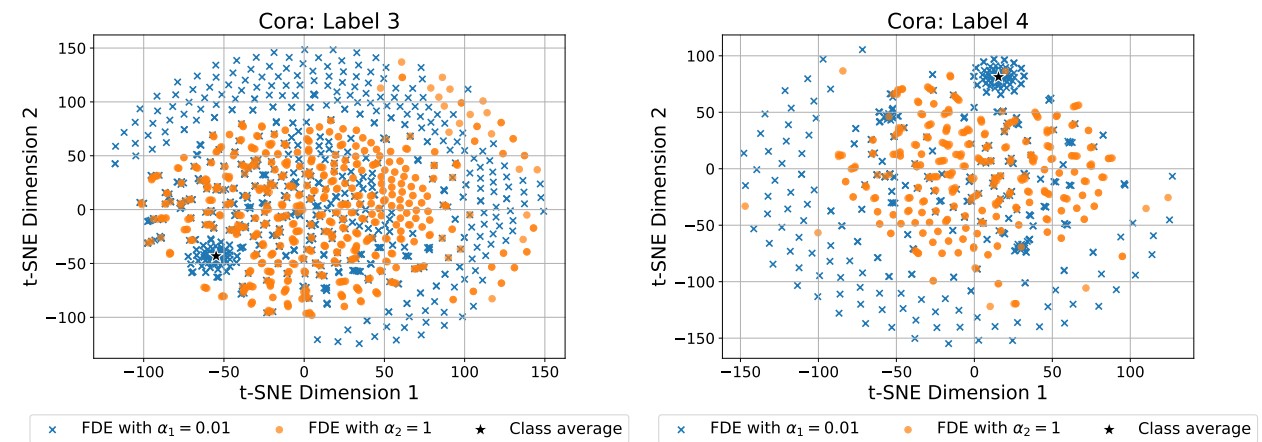

*Figure 8.* Cora: labels 3 and 4

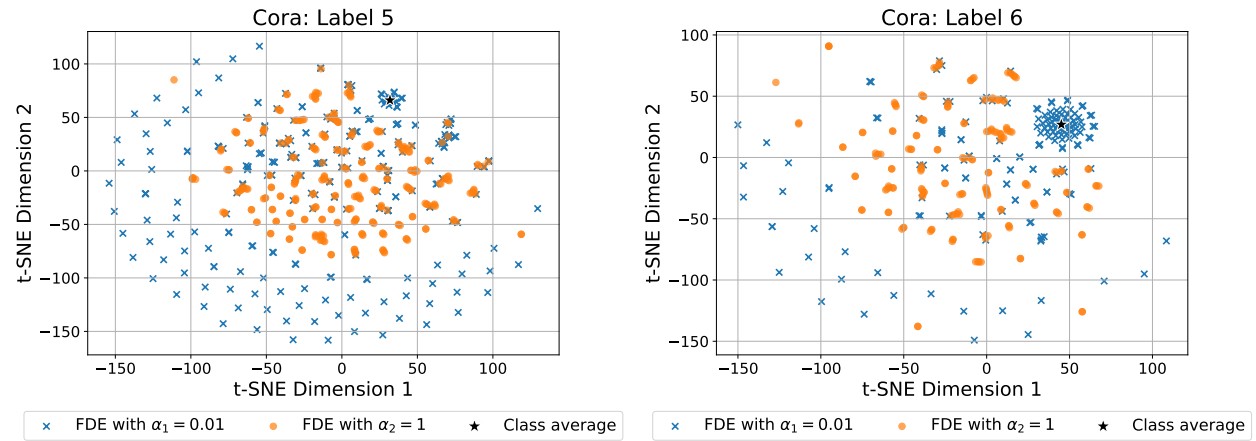

*Figure 9.* Cora: labels 5 and 6

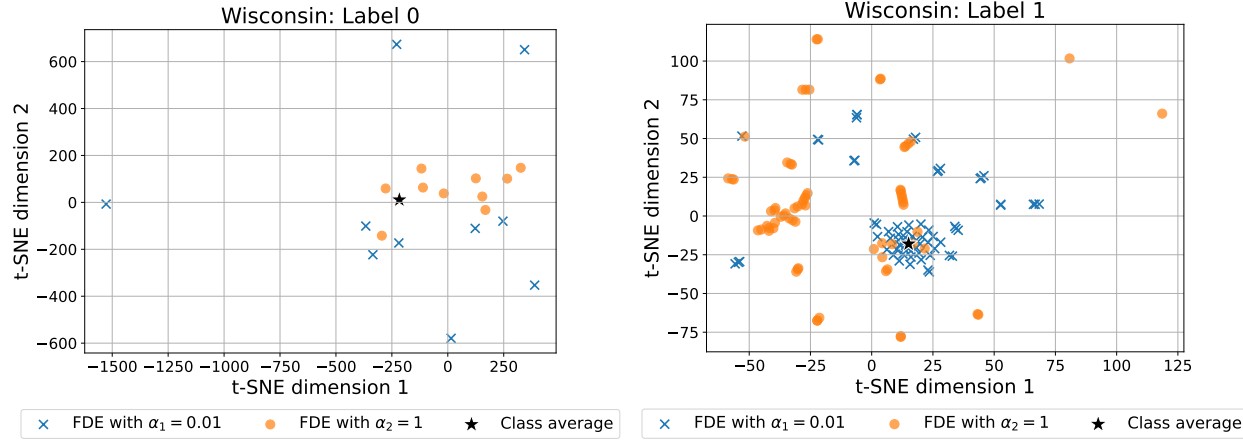

*Figure 10.* Wisconsin: labels 0 and 1

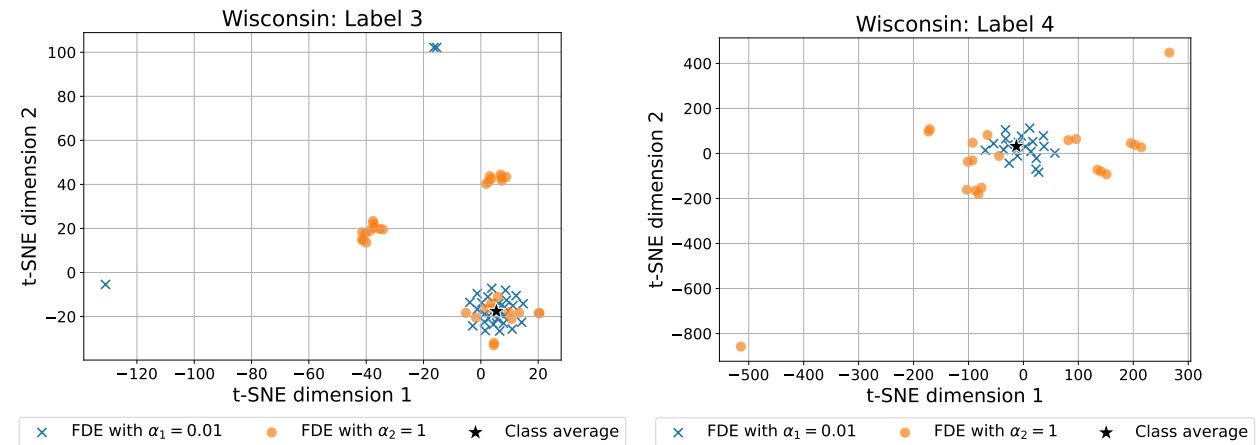

*Figure 11.* Wisconsin: labels 3 and 4

## G.2. Node Classification Results for Different Diffusion Equations

We also report node classification results using alternative choices for $\mathcal{F}$ in (2), such as CDE (5) and GREAD (8). The results, presented in Table 9, demonstrate the generalization and flexibility of FD-GCL.

*Table 9.* Node classification results (%) across different models in FD-GCL. OOM refers to out of memory.

|  | Model | Cora | Wisconsin | Squirrel | Roman |
|---|---|---|---|---|---|
|  | GRAND | 84.20±0.22 | 79.22±5.13 | 64.92±1.46 | 72.56±0.63 |
| FD-GCL | CDE | 71.79±0.14 | 74.31±4.42 | OOM | OOM |
|  | GREAD | 80.36±0.15 | 73.73±4.74 | 65.37±2.00 | 70.97±0.51 |

## G.3. Graph Classification Results

Most existing works on heterophilic graphs primarily address node-level tasks, such as node classification, making empirical evaluation on graph-level tasks less straightforward. To adapt to graph classification, we employ a non-parametric graph pooling (readout) function, such as MeanPooling, to derive graph-level representations. We evaluate the performance of our method on two widely-used graph classification benchmarks: Proteins and DD. The results, summarized in Table 10, demonstrate that our FD-GCL framework is also effective for graph classification, delivering competitive performance compared to baseline methods.

*Table 10.* Graph classification results (%). The best and the second-best result under each dataset are highlighted in **red** and **blue**, respectively.

| Method | Proteins | DD |
|---|---|---|
| InfoGraph(Sun et al., 2019) | 74.44±0.40 | 72.85±1.70 |
| MVGRL(Hassani & Khasahmadi, 2020b) | 74.02±0.30 | 75.20±0.40 |
| GraphCL(You et al., 2020) | 74.39±0.45 | **78.62±0.40** |
| JOAO(You et al., 2021) | 74.55±0.41 | 77.32±0.54 |
| JOAO2(You et al., 2021) | 75.35±0.09 | 77.40±1.11 |
| SimGRACE(Xia et al., 2022) | 75.35±0.09 | 77.44±1.11 |
| DRGCL(Ji et al., 2024) | 75.20±0.60 | 78.40±0.70 |
| CI-GCL (Tan et al., 2024) | **76.50±0.10** | **79.63±0.30** |
| FD-GCL | **75.40±0.28** | 78.53±0.36 |

