# OpenReview forum: "Simple Graph Contrastive Learning via Fractional-order Neural Diffusion Networks"
_ICML.cc/2025/Conference — Submitted to ICML 2025_

### Official Review · Reviewer_xZCw · 2025-02-24

**Overall Recommendation:** 2

**Summary:**

This paper introduces a novel augmentation-free GCL framework. Unlike traditional GCL methods that rely on complex augmentations or negative sampling, this framework uses Fractional Differential Equations to generate different feature views.

**Claims And Evidence:**

The experimental results demonstrate competitive performance, however, some claims need theoretical justification and empirical validation:
1. The authors claim that the main contribution is introducing FDE-based graph contrastive learning. However, the background on FDE is not well-developed. Specifically, why is FDE a better choice than ODE for GCL? Has FDE been applied to GNNs in previous works?
2. While t-SNE and PCA visualizations suggest improved representation learning, they don't prove that FD-GCL mitigates dimensional collapse better than existing methods. It needs a stronger theoretical analysis.

**Essential References Not Discussed:**

The paper does not compare it to some works in negative-free contrastive learning:
[1] Xia, Jun, et al. "Simgrace: A simple framework for graph contrastive learning without data augmentation." Proceedings of the ACM web conference 2022. 2022.
[2] Thakoor, Shantanu, et al. "Large-Scale Representation Learning on Graphs via Bootstrapping." International Conference on Learning Representations.

**Experimental Designs Or Analyses:**

The experiments are well-structured, but there is a concern regarding the completeness:

In the ablation study, the choice of different 𝛼 values has a significant impact on model performance. However, the paper only reports results for a limited set of 𝛼 values, raising concerns about whether different combinations might yield different outcomes. It remains unclear whether the selected 𝛼 values are optimal across all datasets or if dataset-specific tuning is necessary. Without a more comprehensive exploration of 𝛼 variations, the generalizability of the findings is uncertain. A more systematic analysis, testing a broader range of 𝛼 combinations across multiple datasets, would strengthen the experimental validity.

**Methods And Evaluation Criteria:**

The proposed method and evaluation criteria are reasonable, but there is a concern about whether the step size of diffusion significantly impacts the final performance. It would be useful to test the effect of different diffusion depths (T).

**Other Comments Or Suggestions:**

Please refer to above all parts.

**Other Strengths And Weaknesses:**

Please see above all parts.

**Questions For Authors:**

My questions from all the above sections are summarized here:
1. The authors claim that the main contribution is introducing FDE-based graph contrastive learning. However, the background on FDE is not well-developed. Specifically, why is FDE a better choice than ODE for GCL? Has FDE been applied to GNNs in previous works?
2. While t-SNE and PCA visualizations suggest improved representation learning, they don't prove that FD-GCL mitigates dimensional collapse better than existing methods. It needs a stronger theoretical analysis.
3. The proposed method and evaluation criteria are reasonable, but there is a concern about whether the step size of diffusion significantly impacts the final performance. It would be useful to test the effect of different diffusion depths (T).
4. The paper presents theoretical claims regarding the role of FDEs in generating diverse feature views for contrastive learning. The key theoretical argument is that varying fractional order 𝛼 allows the model to control local vs. global feature mixing, thereby improving representation learning. However, the mathematical justification for this claim is intuitive, so the authors should provide formal derivations or proofs.
5. In the ablation study, it remains unclear whether the selected 𝛼 values are optimal across all datasets or if dataset-specific tuning is necessary. Without a more comprehensive exploration of 𝛼 variations, the generalizability of the findings is uncertain. A more systematic analysis, testing a broader range of 𝛼 combinations across multiple datasets, would strengthen the experimental validity.
6. The paper does not compare it to some works in negative-free contrastive learning:
[1] Xia, Jun, et al. "Simgrace: A simple framework for graph contrastive learning without data augmentation." Proceedings of the ACM web conference 2022. 2022.
[2] Thakoor, Shantanu, et al. "Large-Scale Representation Learning on Graphs via Bootstrapping." International Conference on Learning Representations.

**Relation To Broader Scientific Literature:**

The paper situates itself within the broader literature on GCL, specifically in the context of augmentation-free contrastive learning and graph diffusion models. Its primary contribution—leveraging FDEs to control local vs. global feature mixing—draws connections to existing work on graph diffusion models based on ODEs. However, while ODE-based methods have been well studied, the application of FDEs to GCL is relatively novel.

**Theoretical Claims:**

The paper presents theoretical claims regarding the role of FDEs in generating diverse feature views for contrastive learning. The key theoretical argument is that varying fractional order 𝛼 allows the model to control local vs. global feature mixing, thereby improving representation learning. However, the mathematical justification for this claim is intuitive, so the authors should provide formal derivations or proofs.

---

> ### Author Rebuttal · Authors · 2025-03-29
>
> Thank you for the insightful comments and suggestions.
>
> **W1**. Why FDE for GCL.
>
> A core principle of GCL is to generate diverse views, with novelty in how they are constructed. FD-GCL uses neural diffusion-based encoders governed by FDEs, where fractional order $\alpha$ controls diffusion scale—enabling views with varying locality/globality. FDEs generalize ODEs ($\alpha=1$) but offer greater flexibility. Fixed $\alpha=1$ yields less diverse views, weakening contrastive effect. As a generalization, FDEs perform no worse than ODEs while enabling richer view generation, making them a better choice. See **Appendix C and D** for details on FDEs and ODEs.
>
> Theoretically, **Thm 2** (formal version of Thm 1) proves that embeddings generated with different $\alpha$ are provably distinct, with contrast increasing as differences in $\alpha$ grows. Empirical results (**Table R2 for Reviewer ttop**) further confirm FDE's advantage over ODE-based methods (GRAND, GraphCON) for all datasets.
>
> Additionally, this is the **first** work to apply FDE-based diffusion to GCL, offering a novel and flexible mechanism for view generation via tunable diffusion dynamics, despite prior usages of FDEs in GNNs [R1,R2].
>
> [R1] Kang. et al, Unleashing the potential of fractional calculus in graph neural networks with FROND, ICLR, 2024.
>
> [R2] Zhao. et al, Distributed-order fractional graph operating network, Neuips, 2024.
>
> **W2**. FD-GCL mitigates dimensional collapse
>
> Dimension collapse refers to features being confined to a low-dimensional subspace, which may harm CL performance, though low-dimensional features are not inherently poor. Thus, to fairly compare CL models, model performance remains the primary comparison metric for different datasets. Nevertheless, we have provided evidence that FG-GCL can mitigate dimension collapse. Theoretically **(Thm 2 cf. Appendix E)**, we have shown that for small $\alpha$, the generated features tend to belong to a space spanned by a relatively large amount vectors in the spectral domain, which is empirically supported by the PCA visualization (a set of collapsed features should resemble a delta function). Refer to [Fig.(link)](https://limewire.com/d/nubcZ#8VF9FURNHF) for the PCA visualization of PolyGCL (i.e., low-pass and high-pass spectral views), indicates that the number of significant PCA components for FD-GCL (Fig. 2 in our paper) is comparable to that of PolyGCL.
>
> **W3**. The effect of different diffusion depths ($T$)
>
> As rigorously discussed in **Appendix E**, increasing diffusion depth $T$ generally enhances view diversity. We empirically evaluate the effect of varying $T$ on both homophilic and heterophilic datasets in **Table R5**, showing that larger $T$ values consistently lead to improved performance. The values $T$ for each dataset are reported in **Table 8 in Appendix F.3**.
>
> **Table R5. Classification accuracy w.r.t $T$**
>
> |T|5|10|15|20
> |-|-|-|-|-
> |Cora|81.50|82.97|83.68|84.68
> |Ogbn-arxiv|63.77|66.46|66.48|66.18
> |Squirrel|40.67|40.85|57.77|51.08
> |Chameleon|60.65|60.74|70.87|73.18
> |Cornell|60.97|61.08|61.81|68.38
> |Wisconsin|71.57|71.96|73.92|79.02
>
> **W4**. Mathematical analysis of FDEs
>
> This claim is theoretically supported by **Thm 2 (cf. Appendix E)**, which proves that embeddings generated by FDEs with different fractional orders are provably distinct, with the contrast between them increasing as the difference in $\alpha$ grows. The rigorous mathematical analysis can be found in Appendix E. The relation between Theorem 1 and the claim in the comment has been explained in Sec.4.2 (2nd paragraph of ''distinct views''). Intuitively, for large $\alpha$, we have shown that the spectral are more concentrated on low-frequency components. It is well-known in GSP that generally, a low-frequency signal lacks variation across the entire graph and thus it represents a ''global view''. In addition to the theoretical justification, this claim is also empirically validated. As shown in **Fig. 1 and Appendix G.1**, node features generated by two FDE encoders with different fractional orders exhibit clearly distinct characteristics. Specifically, a smaller fractional order (e.g., $\alpha=0.01$) leads to embeddings with a concentrated core, whereas a larger order (e.g., $\alpha=1$) yields features that are more evenly distributed across the space.
>
> **W5**.  Tuning of $\alpha_{1}$ and $\alpha_{2}$
>
> See **Reviewer ttop's W4**.
>
> **W6**.  Lack comparisons with two CL works (e.g., Simgrace and BGRL)
>
> Please note that the requested comparisons have already been included: SimGRACE results are in **Table 10**, and BGRL comparisons are in **Table 1 and 2**. Although SimGRACE is specifically designed for graph classification, FD-GCL achieves comparable results on this task. Moreover, FD-GCL surpasses BGRL on node classification across both homophilic and heterophilic datasets, with notable gains on heterophilic datasets (e.g., 28% on Squirrel/Wisconsin, 26% on Texas/Cornell, 5% on Actor/Roman/Arxiv-year).

---

### Official Review · Reviewer_ZHMy · 2025-03-12

**Overall Recommendation:** 2

**Summary:**

This paper proposed a simple and effective augmentation-free graph contrastive learning framework, which uses Fractional Differential Equations induced graph neural diffusion models . By varying the order parameter,  this method generates diverse views that capture both local and global graph information, eliminating the need for both complex augmentations and negative samples. It achieves state-of-the-art
performance across diverse datasets.

**Claims And Evidence:**

The claims are well-motivated and largely supported by theoretical and empirical evidence.

**Essential References Not Discussed:**

Most related and foundational works are well-cited and discussed, but some latest work on unsupervised graph contrastive learning should also be considered.
[1] LOHA: Direct Graph Spectral Contrastive Learning Between Low-pass and High-pass Views.

**Experimental Designs Or Analyses:**

The experimental designs are reasonable and complete.

**Methods And Evaluation Criteria:**

The success of an augmentation-free approach hinges on two factors: (a) the ability of the encoders to generate high-quality feature embeddings, and (b) the capability of contrasting encoders to produce distinct views of the same input. To address these requirements, this paper propose a novel GCL framework that utilizes neural diffusion-based encoders to generate contrasting views of node features. The proposed methods and evaluation criteria are appropriate and well-aligned with the paper’s goals.

**Other Comments Or Suggestions:**

No, see weaknesses.

**Other Strengths And Weaknesses:**

Strengths:

S1: The paper is well-written and easy to understand.The proposed framework innovatively integrates  Fractional Differential Equations induced graph neural diffusion models with different order parameter as encoders to obtain contrastive views, offering a meaningful solution with practical impact.

S2:  The evaluation of the proposed method is comprehensive, experiments include performance comparisons and more visualization results. Especially in the process of method design, the rationality of the method is fully verified through rich visualization results.

S3: Releasing well-organized source code, allowing reproduction of reported results on the datasets provided.

Weaknesses:

W1: The contribution of this study is limited and insufficient to meet the standards of ICML.  This is mainly because many key technologies are based on existing works. For example,  the fractional-order differential operator $D_{\alpha}^t$ , a cricial technology, is designed by Kang et al. (2024) .

W2: As shown in Theorem 1, different order parameters will result in the augmented views being as distinct as possible. But why is this beneficial for contrastive learning, lacking sufficient theoretical analysis. And, as shown in Figure 3, why can the discrimination of the encoder be enhanced.

W3: For S1 in Section 4.3, the operator  $\mathbf Y_l = \mathbf X \mathbf W_l$  typically increases the feature dimension. What is the purpose of increasing feature dimensions?

W4: Is there a clear pattern in parameters selection of $\alpha_1$ and $\alpha_2$ for homogeneous and heterogeneous graphs. Since parameters $\alpha_1$ and $\alpha_2$ are very important, providing theoretical selection guidance would improve the quality of this paper.

**Questions For Authors:**

Please see the Weakness part above.

**Relation To Broader Scientific Literature:**

The work situates itself at the intersection of graph contrastive learning and graph representation learning. The addressed challenge is important for graph contrastive learning, which makes sense.

**Theoretical Claims:**

I checked all theoretical claims, including proofs in the main paper.

---

> ### Author Rebuttal · Authors · 2025-03-29
>
> Thank you for the insightful comments and suggestions.
>
> **W1**. The novelty of FD-GCL
>
> A general guiding principle for GCL is to generate views from diverse perspectives, with *novelty lies in how these views are generated*. For example, PolyGCL uses polynomial filters for low-pass and high-pass spectral views, while bandpass graph filters are well studied in GSP and GNN. Analogously, the core novelty of FD-GCL is not merely replacing components in existing augmentation-free or negative-free pipelines, but introducing a new perspective for encoder design: generating distinct views via diffusion dynamics, which aligns with GCL's core principle.
>
> FD-GCL uses neural diffusion-based encoders governed by FDEs, where the fractional order $\alpha$ controls the locality/globality of features. By using different $\alpha$ values, FD-GCL produces views with varying diffusion scales. To our knowledge, this is **the first work** to apply diffusion dynamics via FDEs for contrastive learning, offering a novel and flexible *mechanism for view generation through tunable diffusion rates*.
>
> Another technical novelty is the rigorous analysis in **Thm 2** (the formal version of Thm 1), which proves that embeddings generated by FDEs with different fractional orders are provably distinct, and that the contrast between them increases as the difference in $\alpha$ values becomes larger. This is the **first** formal mathematical analysis of this property in GCL, despite the established use of FDEs in GNNs [R1,R2]. This theoretical insight is further supported by numerical evidence in **Figure 1 and Appendix G.1**, where we examine the FDEs under two widely separated fractional orders ($\alpha_{1}=0.01$ and $\alpha_{2}=1$). The results clearly show distinct feature distributions: a small $\alpha$ leads to embeddings with a highly concentrated core, while a large $\alpha$ produces more evenly spread features.
>
> Moreover, unlike BGRL, CCA-SSG, GraphACL and PolyGCL, which rely on either data augmentation or negative sampling, **FD-GCL requires neither**, simplifying usage. AFGRL shares this simplicity but is limited to homophilic datasets. In contrast, FD-GCL is straightforward and effective on both homophilic and heterophilic settings (Table R3).
>
> **Table R3. Comparison with state-of-the-art GCL methods**
> |Method|Argumentation-free| Without negative sampling|Homophilic|Heterophilic|
> |-|-|-|-|-|
> |BGRL| ✘|✔|✔|✔|
> |CCA-SSG|✘|✔|✔|✔|
> |GraphACL|✔|✘|✔|✔|
> |PolyGCL|✔|✘|✔|✔|
> |AFGRL|✔|✔|✔|✘|
> |FD-GCL|✔|✔|✔|✔|
>
> [R1] Kang. et al, Unleashing the potential of fractional calculus in graph neural networks with FROND, ICLR 2024.
>
> [R2] Zhao. et al, Distributed-order fractional graph operating network, NeurIPS 2024.
>
> **W2.** Lack of theoretical analysis & Fig. 3 explanation
>
> The core principle of GCL is to design encoders that generate distinct yet meaningful views. FD-GCL introduces a novel view-generation mechanism via neural diffusion governed by FDEs, where fractional order $\alpha$ controls feature locality versus globality across continuous scales. This mechanism is theoretically grounded, shown in **Theorem 2 (cf. Appendix E)**. We refer to response to **W1** for more details.
>
> Fig. 3 shows the unsupervised clustering capability of FDE-based encoders. For each class $c$, we measure clustering quality using the discrimination ratio $r_c = d_c^{\mathrm{inter}}(\text{intra-class distance})/d_c^{\mathrm{intra}}(\text{inter-class distance})$. The higher the ratio, the better the clustering quality. Each curve (one per class) shows that $r_c$ increases and then stabilizes during training. This trend indicates that the encoder gradually enhances inter-class separation while maintaining intra-class cohesion, which is crucial for classification.
>
> W3. Purpose of increasing feature dimensions
>
> By [R3], increasing feature dimension enhances GCL expressiveness by capturing more complex patterns. We use the operator $Y_l = XW_l$ to project features to a higher dimension, and larger dimensions generally improve performance on all graph types (see Table R4).
>
> [R3] Xiao et al, Simple and asymmetric graph contrastive learning without augmentations, Neuips, 2023.
>
> **Table R4. Classification accuracy w.r.t feature dimension d**
> |d|128|256|512|1024|2048
> |-|-|-|-|-|-
> |Cora|83.27|84.42|83.28|83.13|83.44
> |Citeseer|63.10|64.18|67.78|71.29|73.70
> |Pubmed|77.03|79.93|79.95|80.19|80.57
> |Computer|82.22|85.48|87.81|88.80|90.13
> |Photo|88.66|91.63|92.88|93.41|93.94
> |Squirrel|36.82|44.70|53.29|61.85|64.43
> |Chameleon|59.69|68.09|71.62|72.28|73.53
> |Cornell|58.10|68.64|64.05|68.65|67.58
> |Wisconsin|72.74|76.27|71.76|77.64|77.05
> |Roman|62.73|65.49|68.78|70.56|71.48
>
> **W4**. Tuning of $\alpha_{1}$ and $\alpha_{2}$
>
> Refer to **Reviewer ttop's W4**.
>
> **W5**. Lack of latest work (e.g., LOHA)
>
> Please note that LOHA (AAAI 2025, Jan. 2025) is concurrent work per ICML 2025 policy; thus, direct comparison is not required. Also, the code for LOHA is not public, hindering timely reproduction.

---

> > ### Comment · Reviewer_ZHMy · 2025-04-03
> >
> > Thank you to the authors for the rebuttal. Indeed, the authors have addressed my concerns in some details, but some theoretical analyses are not convincing enough to validate the effectiveness of the proposed method in contrastive augmentation.

---

> > > ### Author Response · Authors · 2025-04-03
> > >
> > > Thank you for the remark. We will be grateful if you could be more specific regarding which particular parts or assumptions of the analysis you believe require further clarification or additional supporting evidence. We will be glad to provide further details if needed and your feedback will be of great help for us to improve our work.

---

### Official Review · Reviewer_ttop · 2025-03-14

**Overall Recommendation:** 2

**Summary:**

This paper proposes Fractional-order Neural Diffusion Networks (FNDN) as a new encoding method for Simple Graph Contrastive Learning (GCL). Unlike augmentation-based GCL approaches that rely on complex data transformations or augmentation-free methods that still require careful encoder design, this work introduces fractional-order differential equations (FDEs) to generate diverse feature views dynamically. The key insight is that the fractional derivative order α controls the extent of local vs. global information captured in node embeddings, allowing different contrastive views without requiring negative samples.

**Claims And Evidence:**

Problematic Claims:

* The paper claims to propose "a novel way of generating contrastive views in GCL", but augmentation-free and negative-free GCL strategies have already been extensively studied (e.g., BGRL, CCA-SSG, AFGRL, GraphACL). The use of FNDN only replaces an existing technique rather than introducing a fundamentally new learning paradigm.
* The claim that FD-GCL is efficient lacks strong empirical backing—while complexity analysis is included, no runtime comparisons on large-scale datasets (e.g., OGB) are provided.
* The claim that FD-GCL naturally avoids feature collapse is not thoroughly analyzed—existing augmentation-free GCL methods often mitigate this issue through architectural modifications or regularization rather than diffusion-based methods.

**Essential References Not Discussed:**

No

**Experimental Designs Or Analyses:**

The experiments are well-designed, though additional scalability validation would be helpful.

**Methods And Evaluation Criteria:**

* The method requires manual tuning of  $\alpha_{1}$ and $\alpha_{2}$, and there is no adaptive strategy for selecting optimal values across different datasets.
* The scalability of FD-GCL is not well-studied—while theoretical complexity analysis is provided, empirical results on large-scale datasets are missing.

**Other Comments Or Suggestions:**

* Clarify novelty compared to augmentation-free baselines: The introduction should explicitly differentiate FD-GCL from existing methods like BGRL, CCA-SSG, GraphACL and explain why fractional diffusion is a meaningful improvement rather than just an alternative.
* Provide runtime and memory efficiency analysis: Adding computation time comparisons against baseline methods would address concerns about efficiency.
* Justify the choice of fractional diffusion over other diffusion strategies: A discussion of how fractional diffusion compares to existing graph diffusion models would strengthen the argument.

**Other Strengths And Weaknesses:**

Strengths:

* Avoids Explicit Data Augmentation and Negative Samples: While augmentation-free GCL is not novel, this method provides an alternative mechanism for generating diverse feature views, avoiding traditional graph perturbation methods.

* Applicability to Both Homophilic and Heterophilic Graphs: The method performs consistently across different graph structures, showing robustness to topology variations.

Weaknesses:

* Incremental Contribution Rather Than Fundamental Innovation: The main idea is not fundamentally novel, as augmentation-free GCL methods already exist. The paper replaces traditional view-generation mechanisms but does not introduce a new learning paradigm.

* Scalability and Efficiency Are Not Demonstrated: The method relies on fractional-order diffusion, which may introduce additional computational overhead. However, the paper does not include a runtime analysis or test on large-scale datasets (e.g., OGB). This raises concerns about its practical applicability.

* Manual Hyperparameter Selection: The selection of $\alpha_{1}$ and $\alpha_{2}$ is manual, making the method less adaptive to different datasets. There is no guidance or automated strategy for tuning these parameters.

*  No Discussion on Alternative Diffusion Models: The paper focuses solely on fractional diffusion but does not compare it to other graph diffusion techniques (e.g., heat diffusion, GRAND, GraphCON). A comparison would strengthen the justification for choosing fractional-order diffusion over other approaches.

**Questions For Authors:**

Q1: How does fractional-order diffusion compare to other graph diffusion methods (e.g., heat diffusion, Personalized PageRank, GRAND, GraphCON)?

Q2: How does FD-GCL prevent feature collapse compared to existing augmentation-free GCL methods?

**Relation To Broader Scientific Literature:**

* Graph Contrastive Learning (GCL): The method aligns with augmentation-free GCL techniques (e.g., BGRL, CCA-SSG, GraphACL), but instead of using architectural tricks or spectral filtering, it leverages fractional-order diffusion.
* Graph Neural Diffusion Models: Related to ODE-based graph diffusion methods (GRAND, GraphCON) but extends them to fractional derivatives.
* Graph Signal Processing (GSP): The use of fractional diffusion has connections to spectral graph theory, though it has been studied before in graph signal reconstruction and filtering.

While the paper connects well to existing literature, its contribution is incremental rather than groundbreaking.

**Theoretical Claims:**

Theorem 1 provides a solid spectral analysis of how different fractional orders affect node embeddings, and it aligns with established graph signal processing principles.

The derivation of the diffusion process is technically sound, but it does not introduce a fundamentally new theoretical framework—fractional diffusion has been previously studied in graph signal processing and neural PDE-based models.

No major mathematical flaws were found, but the impact of this analysis on contrastive learning remains unclear beyond providing an alternative view-generation method.

---

> ### Author Rebuttal · Authors · 2025-03-29
>
> Thank you for the insightful comments and suggestions.
>
> **W1**. The novelty of FD-GCL
>
> FD-GCL's novelty is not merely replacing components of existing augmentation-free or negative-free pipelines, but introducing a new perspective for encoder design: generating distinct views via diffusion dynamics, aligning with GCL's core principle. **See W1 of Reviewer ZHMy on both architectural and theoretical novelty**.
>
> **W2**. Scalability & efficiency of FD-GCL
>
> Scalability and efficiency are illustrated by training and running time comparisons on the large-scale Ogbn-arxiv dataset, shown in **Table 4** in the paper and Table R1, respectively. The results confirm that FD-GCL maintains competitive training and running times, highlighting its practical efficiency. Additional evaluations on other large-scale Roman-empire and Arxiv-year datasets (**Table 1 and 2**) further support its scalability, with dataset sizes comparable to benchmarks like PolyGCL.
>
> **Table R1. Testing time (sec)**. OOM refers to out of memory on an NVIDIA RTX A5000 GPU (24GB) during training.
>
> |Method|Cora|Wisconsin|Ogbn-arxiv|
> |-|-|-|-|
> |GraphACL|13.29|38.69|14.59|
> |PolyGCL|7.46|9.73|OOM|
> |FD-GCL|2.96|2.93|14.88|
>
> **W3**. How FD-GCL mitigates dimensional and feature collapse
>
> Dimension collapse refers to embeddings being confined to a low-dimensional subspace. In **Theorem 2 (cf. Appendix E)** , we theoretically prove that using a small fractional order $\alpha_{1}$ mitigates this issue by reducing energy concentration in the spectral domain. Specifically, it shows that $\mathbf{Z}_{\alpha_1}(t) $ is less energy concentrated in the spectral domain,
>
> i.e., it has a decomposition $\sum_{1\leq i\leq N} c_i\mathbf{u}_i$ with many large $|c_i|$, meaning the embeddings span a higher-dimensional subspace. This is further supported by PCA results (Fig. 2), where smaller $\alpha$ values preserve significantly more principal components.
>
> On the other hand, view collapse means generated views converge to similar representations. We interpret the reviewer is referring to view collapse. While existing augmentation-free GCL methods typically rely on architectural changes or explicit regularization to address feature collapse, FD-GCL adopts a fundamentally different approach by leveraging fractional diffusion dynamics. We mitigate this collapse with a regularized cosmean loss (i.e., $\mathcal{L}(\mathbf{Z}_1,\mathbf{Z}_2)= \mathcal{L}_0(\mathbf{Z}_1,\mathbf{Z}_2)+\eta |\langle \mathbf{c}_1, \mathbf{c}_2 \rangle|$) that includes a penalty term on the angle between the dominant directions $\mathbf{c}_1$ and $\mathbf{c}_2$ of embeddings $\mathbf{Z}_1$ and $\mathbf{Z}_2$. This encourages diversity between the views without relying on negative samples. This approach is possible as we have observed that features from each view of FD-GCL tend to have a pronounced dominant component. As demonstrated in **Fig. 5 and Fig. 6 in Appendix F.4**, this regularization ensures stable performance across training epochs. Combined with our theoretical findings, these results provide strong evidence that FD-GCL effectively avoids feature collapse through its diffusion-based framework.
>
> **W4**. Tuning of $\alpha_1$ and $\alpha_2$
>
> We refer the reviewers to **Appendix F.3**. Both $\alpha_{1}$ and $\alpha_{2}$ are tunable within the range $(0,1]$. Motivated by the theoretical insights in Thm 1 (or Thm 2), which suggest that a larger difference between $\alpha_2$ and $\alpha_1$ enhances the contrast between views, we adopt a simple yet effective strategy: we fix $\alpha_{2}=1$ to maintain a consistent global view and tune $\alpha_{1}$ over the range $(0,1]$ using a grid search. This approach is guided by our analysis and provides a practical, computationally efficient solution for tuning $\alpha_l$. The corresponding values $\alpha_{1}$ and $\alpha_{2}$ for each dataset are reported in **Table 8 in Appendix F.3**. We recognize that more adaptive or data-driven methods for selecting fractional orders are promising. To this end, in our future GCL work, we may adopt variable-order fractional derivatives where the derivative orders depend on hidden features.
>
> **W5**. Compare with alternative diffusion models.
>
> Note that the fractional diffusion model in FD-GCL generalizes other graph diffusion techniques (e.g., GRAND (cf. (3) in Appendix C) and GraphCON (cf. (4) in Appendix C)) by setting $\alpha_{1}=\alpha_{2}=1$. Table R2 shows this fixed setting yields poorer results on both homophilic and heterophilic datasets, since it produces less distinct views, weakening the contrastive effect. This highlights the advantage of fractional-order diffusion.
>
> **Table R2. Classification accuracy on different graph diffusion models**.
>
> |Method|GRAND|GraphCON|FD-GCL|
> |-|-|-|-|
> |Cora|78.09±0.19|76.50±0.10|84.27±0.27|
> |Ogbn-arxiv|66.37±0.13|OOM|70.46±0.13|
> |Crocodile|63.57±1.01|67.79±0.60|68.99±0.66|
> |Wisconsin|61.57±6.21|62.35±5.74|79.22±5.13|
> |Arxiv-year|47.08±0.15|43.93±0.13|47.22±0.13|

---

> > ### Comment · Reviewer_ttop · 2025-04-02
> >
> > Thank you to the authors for the rebuttal. I will keep my original score.

---

> > > ### Author Response · Authors · 2025-04-02
> > >
> > > Thank you for reading our rebuttal. Could you please let us know whether it has resolved you concerns regarding the paper? If you have any additional questions, we would be happy to provide further clarification if needed.

---

### Decision · Program_Chairs · 2025-05-01

**Decision:**

Reject

**Comment:**

This paper studies the problem of graph contrastive learning and proposes a new approach based on graph neural diffusion models. The model utilizes Fractional Differential Equations (FDE) to extract representations from diverse views. These views are then incorporated for graph contrastive learning.

The reviewers hold a unanimous rejection after rebuttal. The reviewers have concerns on unconvincing theoretical analyses, novelty compared with existing augmentation-free GCL methods, limited contributions, and the motivation of introducing GCL, and missing related works.

Overall, the paper stands below the borderline of prestigious ICML. I'm inclined to reject the paper and encourage the authors to revise the paper based on reviewers' comments for future venues.